# Split Gibbs Discrete Diffusion Posterior Sampling

**Wenda Chu**[1]     **Zihui Wu**[1]     **Yifan Chen**[2]     **Yang Song**[3]     **Yisong Yue**[1]

[1]California Institute of Technology     [2]New York University     [3]OpenAI

## Abstract

We study the problem of posterior sampling in discrete-state spaces using discrete diffusion models. While posterior sampling methods for continuous diffusion models have achieved remarkable progress, analogous methods for discrete diffusion models remain challenging. In this work, we introduce a principled plug-and-play discrete diffusion posterior sampling algorithm based on split Gibbs sampling, which we call SGDD. Our algorithm enables reward-guided generation and solving inverse problems in discrete-state spaces. We demonstrate the convergence of SGDD to the target posterior distribution and verify this through controlled experiments on synthetic benchmarks. Our method enjoys state-of-the-art posterior sampling performance on a range of benchmarks for discrete data, including DNA sequence design, discrete image inverse problems, and music infilling, achieving more than $30\%$ improved performance compared to existing baselines.

## 1 Introduction

We study the problem of posterior sampling in finite spaces with discrete diffusion models. Given a pretrained discrete diffusion model [35, 1, 4, 29] that models the prior distribution $p(\mathbf{x})$, our goal is to sample from the posterior $p(\mathbf{x}|\mathbf{y})$ in a plug-and-play fashion, where $\mathbf{y}$ represents a guiding signal of interest. This setting encompasses both solving inverse problems (e.g., if $\mathbf{y}$ is an incomplete measurement of $\mathbf{x}$), and conditional or guided generation (e.g., sampling from $p(\mathbf{x}) \exp(\beta r(\mathbf{x}))/Z$ according to a reward function $r(\cdot)$).

Existing works on diffusion posterior sampling [7, 37, 30, 51] primarily focus on continuous diffusion models operating in the Euclidean space for $\mathbf{x}$. These methods have achieved remarkable success in applications including natural image restorations [24, 7, 57], medical imaging [20, 42, 6], and various scientific inverse problems [14, 51, 55]. However, posterior sampling with discrete diffusion models in discrete-state spaces remains a challenging problem, mainly due to not having well-defined gradients of the likelihood function $p(\mathbf{y}|\mathbf{x})$. Initial work to address this gap for discrete diffusion models relies on approximating value functions (similar to reinforcement learning) for derivative-free sampling [27], which can be difficult to tune for good performance, particularly when the guidance $\mathbf{y}$ is complicated.

In this paper, we develop a rigorous plug-and-play method based on the principle of split Gibbs sampling [47], which we refer to as SGDD. Our method addresses the posterior sampling problem $\mathbf{x} \sim p(\mathbf{x}|\mathbf{y})$ by introducing an auxiliary variable $\mathbf{z}$ along with a regularization potential function $D(\mathbf{x}, \mathbf{z}; \eta)$ to relax the original problem. The relaxed formulation enables efficient sampling via Gibbs sampling, which alternates between a likelihood sampling step and a prior sampling step using a pretrained discrete diffusion model. As regularization increases, both variables $\mathbf{x}$ and $\mathbf{z}$ converge to the target posterior distribution. Our method generalizes split Gibbs samplers with $\ell_2$ regularizers to any potential functions with certain limiting conditions, making it compatible with discrete diffusion models by carefully designing the potential function. We provide a rigorous stationary guarantee on the convergence of SGDD, with error bounds that account for imperfect score function and discretization errors.

39th Conference on Neural Information Processing Systems (NeurIPS 2025).

We validate SGDD on diverse inverse problems and reward-guided generation problems that involve discrete data. We demonstrate that SGDD converges to the true posterior distribution on synthetic data, that SGDD solves discrete inverse problems accurately and efficiently on discretized image data and monophonic music data, and that SGDD excels at reward-guided sampling on guided DNA generation. Our method demonstrates strong sampling quality in all experiments. For instance, we achieve a 42% higher median activity in enhancer DNA design, an 8.36 dB improvement of PSNR values in solving the XOR inverse problem on MNIST, and over $2\times$ smaller Hellinger distance in music infilling, compared to existing methods.

## 2 Preliminaries

### 2.1 Discrete Diffusion Models

Diffusion models [39, 40, 18, 41, 22, 23] generate data $\mathbf{x} \in \mathcal{X}$ by reversing a predefined forward process. When $\mathcal{X} \in \mathbb{R}^n$ is a Euclidean space, the forward process is typically defined to be a stochastic differential equation [22]: $\mathrm{d}\mathbf{x}_t = \sqrt{2\dot{\sigma}_t\sigma_t}\mathrm{d}\mathbf{w}_t$, where $\mathbf{w}_t$ is a standard Wiener process and $\{\sigma_t\}_{t=0}^T$ is a predefined noise schedule. This diffusion process is reversible once the score function $s(\mathbf{x}_t, t) := \nabla_{\mathbf{x}_t} \log p(\mathbf{x}_t; \sigma_t)$ is learned. Starting from $\mathbf{x}_T \sim \mathcal{N}(0, \sigma_T^2\mathbf{I})$, we generate data following the reverse SDE:

$$\mathrm{d}\mathbf{x}_t = -2\dot{\sigma}_t\sigma_t\nabla_{\mathbf{x}_t} \log p(\mathbf{x}_t; \sigma_t)\mathrm{d}t + \sqrt{2\dot{\sigma}_t\sigma_t}\mathrm{d}\mathbf{w}_t. \tag{1}$$

Recent works on discrete diffusion models [35, 1, 4, 29] extend score-based generative methods from modeling continuous distributions in Euclidean spaces to categorical distributions in discrete-state spaces. Specifically, when the data distribution lies in a finite support $\mathcal{X} = \{1, \ldots, N\}$, one can evolve a family of categorical distributions $p_t$ over $\mathcal{X}$ following a continuous-time Markov chain over the discrete elements,

$$\partial_t p_t = \mathbf{Q}_t^{\text{fw}}p_t, \tag{2}$$

where $p_0 = p_{\text{data}}$ and $\mathbf{Q}_t^{\text{fw}} \in \mathbb{R}^{N \times N}$ are diffusion matrices with a simple stationary distribution. To reverse this continuous-time Markov chain, it suffices to learn the **concrete score** $s(\mathbf{x}; t) := [\frac{p_t(\tilde{\mathbf{x}})}{p_t(\mathbf{x})}]_{\tilde{\mathbf{x}} \neq \mathbf{x}}$, as the reverse process is given by

$$\partial_t p_{T-t} = \mathbf{Q}_t p_{T-t} \tag{3}$$

with $\mathbf{Q}_t^{[i,j]} = \frac{p_{T-t}(\mathbf{x}_i)}{p_{T-t}(\mathbf{x}_j)}\mathbf{Q}_{T-t}^{\text{fw}[j,i]}$ and $\mathbf{Q}_t^{[i,i]} = -\sum_{\mathbf{x}_j \neq \mathbf{x}_i} \mathbf{Q}_t^{[j,i]}$.

For sequential data $\mathbf{x} \in \mathcal{X}^D$, there are $N^D$ states in total. Instead of constructing an exponentially large diffusion matrix, we use a sparse matrix $\mathbf{Q}_t^{\text{fw}}$ that perturbs tokens independently in each dimension [29].

**Example: uniform kernel.** An example of such diffusion matrices is

$$\mathbf{Q}_t^{\text{fw}} = \dot{\sigma}_t\mathbf{Q}^{\text{uniform}} = \dot{\sigma}_t(\mathbf{1}\mathbf{1}^T/N - \mathbf{I}), \tag{4}$$

where $\sigma_t$ is a predefined noise schedule with $\sigma_0 = 0$ and $\sigma_T = \sigma_{\max}$. This uniform kernel transfers any distribution to a uniform distribution as $\sigma \to \infty$. Moreover,

$$p_t = \exp\left(\int_0^t \mathbf{Q}_\tau^{\text{fw}}\mathrm{d}\tau\right)p_0 = \exp(\sigma_t\mathbf{Q}^{\text{uniform}})p_0 = \left(e^{-\sigma_t}\mathbf{I} + (1 - e^{-\sigma_t})\frac{\mathbf{1}\mathbf{1}^T}{N}\right)p_0. \tag{5}$$

When $\mathbf{x} \in \mathcal{X}^D$ is a sequence of length $D$, we have $p_t(\mathbf{x}_t|\mathbf{x}_0) \propto \beta_t^{d(\mathbf{x}_t, \mathbf{x}_0)}(1 - \beta_t)^{D-d(\mathbf{x}_t, \mathbf{x}_0)}$, where $d(\cdot, \cdot)$ is the Hamming distance between two sequences, and $\beta_t = \frac{N-1}{N}(1 - e^{-\sigma_t})$.

### 2.2 Diffusion Posterior Sampling Methods

Diffusion posterior sampling is a class of methods that draw samples from $p(\mathbf{x}|\mathbf{y}) \propto p(\mathbf{x})p(\mathbf{y}|\mathbf{x})$ using a diffusion model trained on the prior $p(\mathbf{x})$. This technique has been applied in two settings with slightly different formulations: solving inverse problems and reward guided generation.

The goal of inverse problems is to reconstruct the underlying data $\mathbf{x}_0$ from its measurements, which are modeled as $\mathbf{y} = \mathbf{G}(\mathbf{x}_0) + \mathbf{n}$, where $\mathbf{G}$ is the forward model, and $\mathbf{n}$ denotes the measurement noise. In this case, the likelihood function is given by $p(\mathbf{y}|\mathbf{x}) = p_{\mathbf{n}}(\mathbf{y} - \mathbf{G}(\mathbf{x}))$, where $p_{\mathbf{n}}$ denotes the noise distribution.

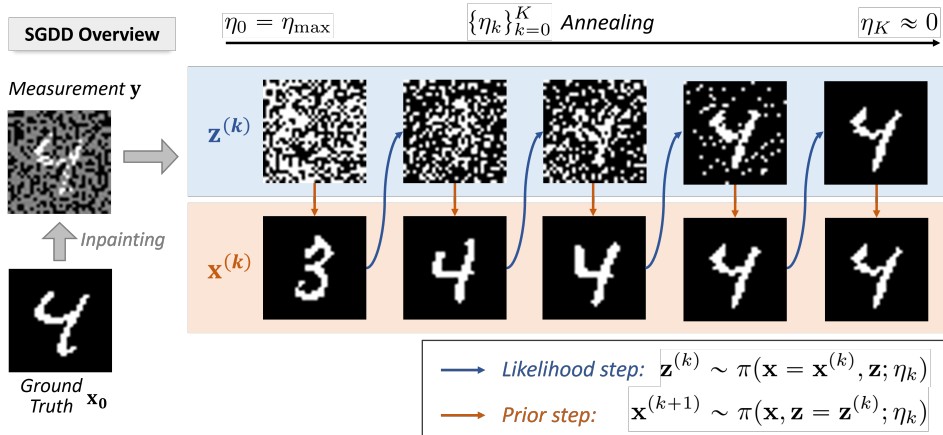

Figure 1: **An illustration of our method on inpainting discretized MNIST.** Our method implements the Split Gibbs sampler, which alternates between likelihood sampling steps and prior sampling steps. Likelihood sampling steps enforce data consistency with regard to measurement $\mathbf{y}$, while prior sampling steps involve denoising uniform noise from $z^{(k)}$ by a pretrained discrete diffusion model starting from noise level $\eta_k$. Two variables converge to a sample from the posterior distribution as the noise level $\eta_k$ reduces to zero.

On the other hand, reward-guided generation [38, 19, 32] focuses on generating samples that achieve higher rewards according to a given reward function $r(\cdot)$. Here, the likelihood is modeled as $p(\mathbf{y}|\mathbf{x}) \propto \exp(\beta r(\mathbf{x}))$, where $\beta > 0$ is a parameter that controls the strength of reward weighting.

Existing diffusion posterior sampling methods leverage (continuous) diffusion models as plug-and-play priors and modify Eq. (1) to generate samples from the posterior distribution $p(\mathbf{x}|\mathbf{y}) \propto p(\mathbf{x})p(\mathbf{y}|\mathbf{x})$. These methods can be broadly classified into four categories [9, 56].

**Guidance-based methods** [7, 38, 37, 24, 49, 55] perform posterior sampling by adding an additional likelihood score term $\nabla_{\mathbf{x}_t} \log p(\mathbf{y}|\mathbf{x}_t)$ to the score function in Eq. (1), and modifies the reverse SDE as: $d\mathbf{x}_t = -2\dot{\sigma}_t\sigma_t(\nabla_{\mathbf{x}_t} \log p(\mathbf{x}_t; \sigma_t) + \nabla_{\mathbf{x}_t} \log p(\mathbf{y}|\mathbf{x}_t))dt + \sqrt{2\dot{\sigma}_t\sigma_t}d\mathbf{w}_t$. The intractable likelihood score term is approximated with various assumptions. However, these methods cannot be directly applied to discrete posterior sampling problems, where $\log p(\mathbf{y}|\mathbf{x})$ is defined only on a finite support and thus does not have such gradient information.

**Sequential Monte Carlo** (SMC) methods sample batches of particles iteratively from a series of distributions, which converge to the posterior distribution in the limit. This class of methods [45, 12, 26, 50, 33, 34, 5] exploits the sequential diffusion sampling process to sample from the posterior distribution. Similar ideas have been applied in Li et al. [27] for reward-guided generation using discrete diffusion models, which rely on approximating a soft value function at each iteration.

**Variational methods** [30, 14] solve inverse problems by approximating the posterior distribution with a parameterized distribution $q_\theta$ and optimizing its Kullback-Leibler (KL) divergence with respect to the posterior. However, how to extend these methods to discrete diffusion models remains largely unexplored.

**Variable splitting** methods [51, 54, 57, 36, 52] decompose posterior sampling into two simpler alternating steps. A representative of this category is the split Gibbs sampler (see Section 3), where the variable $\mathbf{x}_t$ is split into two random variables that both eventually converge to the posterior distribution. These methods are primarily designed for continuous diffusion models with Gaussian kernels, leveraging Langevin dynamics or optimization in Euclidean space as subroutines. Murata et al. [31] extends this category to VQ-diffusion models in the discrete latent space using the Gumbel-softmax reparameterization trick. However, this approach is limited to discrete latent spaces with an underlying continuous embedding due to its reliance on softmax dequantization. In this work, we generalize the split Gibbs sampler to posterior sampling in categorical data by further exploiting the unique properties of discrete diffusion models.

## 3 Method

### 3.1 Split Gibbs for plug-and-play posterior sampling

Suppose we observe $\mathbf{y}$ from a likelihood function $p(\mathbf{y}|\mathbf{x})$ and assume that $\mathbf{x}$ satisfies a prior distribution $p(\mathbf{x})$. Our goal is to sample from the posterior distribution:

$$p(\mathbf{x}|\mathbf{y}) \propto p(\mathbf{y}|\mathbf{x})p(\mathbf{x}) = \exp(-f(\mathbf{x};\mathbf{y}) - g(\mathbf{x})), \tag{6}$$

where $f(\mathbf{x};\mathbf{y}) = -\log p(\mathbf{y}|\mathbf{x})$ and $g(\mathbf{x}) = -\log p(\mathbf{x})$. The challenge in sampling from this distribution arises from the interplay between the prior distribution, modeled by a diffusion model, and the likelihood function.

The Split Gibbs sampler (SGS) relaxes this sampling problem by introducing an auxiliary variable $\mathbf{z}$, allowing sampling from an augmented distribution:

$$\pi(\mathbf{x}, \mathbf{z}; \eta) \propto \exp(-f(\mathbf{z};\mathbf{y}) - g(\mathbf{x}) - D(\mathbf{x}, \mathbf{z}; \eta)), \tag{7}$$

where $D(\mathbf{x}, \mathbf{z}; \eta)$ measures the distance between $\mathbf{x}$ and $\mathbf{z}$, and $\eta > 0$ is a parameter that controls the strength of regularization. While prior works [8, 51, 52] consider $D(\mathbf{x}, \mathbf{z}; \eta) = \frac{\|\mathbf{x}-\mathbf{z}\|_2^2}{2\eta^2}$, the potential function can be generalized to any continuous function $D$, such that $D(\mathbf{x}, \mathbf{z}; \eta) \to \infty$ as $\eta \to 0$ for any $\mathbf{x} \neq \mathbf{z}$. As shown in Appendix A.3, this ensures that both marginal distributions,

$$\pi^X(\mathbf{x}; \eta) := \int \pi(\mathbf{x}, \mathbf{z}; \eta) \mathrm{d}\mathbf{z}, \quad \text{and} \quad \pi^Z(\mathbf{z}; \eta) := \int \pi(\mathbf{x}, \mathbf{z}; \eta) \mathrm{d}\mathbf{x}$$

converges to the posterior distribution $p(\mathbf{x}|\mathbf{y})$. The decoupling of the prior and likelihood in Eq. (7) enables split Gibbs sampling, which alternates between two steps:

1. **Likelihood sampling step:**

$$\mathbf{z}^{(k)} \sim \pi(\mathbf{x} = \mathbf{x}^{(k)}, \mathbf{z}; \eta) \propto \exp(-f(\mathbf{z};\mathbf{y}) - D(\mathbf{x}^{(k)}, \mathbf{z}; \eta)), \tag{8}$$

2. **Prior sampling step:**

$$\mathbf{x}^{(k+1)} \sim \pi(\mathbf{x}, \mathbf{z} = \mathbf{z}^{(k)}; \eta) \propto \exp(-g(\mathbf{x}) - D(\mathbf{x}, \mathbf{z}^{(k)}; \eta)). \tag{9}$$

A key feature of split Gibbs sampling is that it does not rely on knowing gradients of the guidance term $f(\mathbf{z};\mathbf{y})$, which is highly desirable in our setting with discrete data. In this way, split Gibbs is arguably the most natural and simplest framework for developing principled posterior sampling algorithms for discrete diffusion models.

Prior work has studied split Gibbs posterior sampling for continuous diffusion models [8, 51, 52]. These methods specify the regularization potential as $D(\mathbf{x}, \mathbf{z}; \eta) = \frac{\|\mathbf{x}-\mathbf{z}\|_2^2}{2\eta^2}$, which transforms Eq. (9) into a Gaussian denoising problem solvable by a diffusion model. A key challenge, which we tackle, is developing a performant approach for the discrete diffusion setting that is easy to implement, and enjoys rigorous guarantees (i.e., sampling from the correct posterior distribution).

### 3.2 Prior Step with Discrete Diffusion Models

Suppose $p(\mathbf{x})$ is a discrete-state distribution over $\mathcal{X}^D$ modeled by a diffusion process. We consider a discrete diffusion model with uniform transition kernel $\boldsymbol{Q}_t^{\mathrm{fw}} = \frac{1}{N}\mathbf{1}\mathbf{1}^T - \boldsymbol{I}$. To connect split Gibbs sampling to discrete diffusion models, we specify the potential function $D(\mathbf{x}, \mathbf{z}; \eta)$ as:

$$D(\mathbf{x}, \mathbf{z}; \eta) := d(\mathbf{x}, \mathbf{z}) \log \frac{1 + (N-1)e^{-\eta}}{(N-1)(1 - e^{-\eta})} \tag{10}$$

where $d(\mathbf{x}, \mathbf{z})$ denotes the Hamming distance between $\mathbf{x}$ and $\mathbf{z}$. When $\eta \to 0^+$, the regularization potential $D(\mathbf{x}, \mathbf{z}; \eta)$ goes to infinity unless $d(\mathbf{x}, \mathbf{z}) = 0$, ensuring the convergence of marginal distributions to $p(\mathbf{x}|\mathbf{y})$. Given Eq. (10), the prior step from Eq. (9) can be written as

$$\mathbf{x}^{(k+1)} \sim \pi(\mathbf{x}, \mathbf{z} = \mathbf{z}^{(k)}; \eta) \propto p_0(\mathbf{x}) \left( \tilde{\beta}/(1 - \tilde{\beta}) \right)^{d(\mathbf{z}^{(k)}, \mathbf{x})} \tag{11}$$

---

**Algorithm 1** **S**plit **G**ibbs **D**iscrete **D**iffusion Posterior Sampling (SGDD)

---

**Require:** Concrete score model $\mathbf{s}_\theta$, measurement $\mathbf{y}$, noise schedule $\{\eta_k\}_{k=0}^{K-1}$.
Initialize $\mathbf{x}_0 \in \mathcal{X}^d$
**for** $k = 0, \ldots, K - 1$ **do**
  Likelihood step following Eq. (13): $\mathbf{z}^{(k)} \sim \pi(\mathbf{x} = \mathbf{x}^{(k)}, \mathbf{z}; \eta_k)$.
  Prior step following Eq. (11): $\mathbf{x}^{(k+1)} \sim \pi(\mathbf{x}, \mathbf{z} = \mathbf{z}^{(k)}; \eta_k)$
**end for**
**Return** $\mathbf{x}^{(K)}$

---

where $\tilde{\beta} = \frac{N-1}{N}(1 - e^{-\eta})$. On the other hand, the distribution of clean data $\mathbf{x}_0$ conditioned on $\mathbf{x}_t$ is given by:

$$p(\mathbf{x}_0|\mathbf{x}_t) \propto p_0(\mathbf{x}_0)p(\mathbf{x}_t|\mathbf{x}_0) \propto p_0(\mathbf{x}_0)\beta_t^{d(\mathbf{x}_t,\mathbf{x}_0)}(1 - \beta_t)^{D-d(\mathbf{x}_t,\mathbf{x}_0)} \tag{12}$$

where $\beta_t = \frac{N-1}{N}(1 - e^{-\sigma_t})$. Thus, sampling from Eq. (11) is equivalent to unconditional generation from $p(\mathbf{x}_0|\mathbf{x}_t)$ when $\tilde{\beta} = \beta_t$, i.e., $\eta = \sigma_t$. Therefore, we can solve the prior sampling problem by simulating a partial discrete diffusion sampler starting from $\sigma_t = \eta$ and $\mathbf{x}_t = \mathbf{z}^{(k)}$.

### 3.3 Likelihood Sampling Step

With the potential function $D(\mathbf{x}, \mathbf{z}; \eta)$ specified in Section 3.2, at iteration $k$, the likelihood sampling step from Eq. (8) can be written as:

$$\mathbf{z}^{(k)} \sim \pi(\mathbf{x} = \mathbf{x}^{(k)}, \mathbf{z}; \eta) \propto \exp\left(-f(\mathbf{z}; \mathbf{y}) - d(\mathbf{x}, \mathbf{z}) \log \frac{1 + (N-1)e^{-\eta}}{(N-1)(1 - e^{-\eta})}\right). \tag{13}$$

Since the unnormalized probability density function of $\pi(\mathbf{x} = \mathbf{x}^{(k)}, \mathbf{z}; \eta)$ is accessible, we can efficiently sample from Eq. (13) using Markov Chain Monte Carlo methods.

### 3.4 Overall Algorithm

We now summarize the complete SGDD algorithm. Like in Wu et al. [51], our algorithm alternates between likelihood steps and prior steps while employing an annealing schedule $\{\eta_k\}$, which starts at a large $\eta_0$ and gradually decays to $\eta_K \to 0$. This annealing scheme accelerates the mixing time of the Markov chain, which we further provide some theoretical intuition in Appendix B.1; and ensures the convergence of $\pi^X(\mathbf{x}; \eta)$ and $\pi^Z(\mathbf{z}; \eta)$ to $p(\mathbf{x}|\mathbf{y})$ as $\eta \to 0$. We present the complete pseudocode of our method in Algorithm 1 and further explain the choice of the noise schedule in Appendix B.1.

### 3.5 Convergence of Split Gibbs Diffusion Posterior Sampling

We provide theoretical guarantees on the convergence of SGDD. For two probability measures in a finite $\mathcal{X}$, we consider the *Kullback-Leibler (KL) divergence* and the *relative Fisher information (a.k.a. Fisher divergence)*, respectively, as (where we define $f := \mu/\pi$)

$$\mathsf{KL}(\mu\|\pi) := \mathbb{E}_{\mathbf{x}\sim\mu}[\ln\frac{\mu}{\pi}(\mathbf{x})], \quad \mathsf{FI}_{\boldsymbol{Q}}(\mu\|\pi) := \sum_{\mathbf{x}_i, \mathbf{x}_j \in \mathcal{X}} \pi(\mathbf{x}_i)\boldsymbol{Q}^{[j,i]}\left(f(\mathbf{x}_j) - f(\mathbf{x}_i) - f(\mathbf{x}_i)\log\frac{f(\mathbf{x}_j)}{f(\mathbf{x}_i)}\right).$$

Both divergences are nonnegative and equal to zero if and only if $\mu = \pi$, when $\boldsymbol{Q}$ is irreducible. Fisher information has been widely used to analyze the convergence of sampling algorithms in continuous spaces due to its inherent connection to KL divergence [2, 43, 51]. In the continuous case, the Fisher information can be written as a quadratic form $\mathsf{FI}(\mu\|\pi) = \mathbb{E}_\mu\|\nabla\log(\mu/\pi)\|^2$. However, it does not have an analogous form in finite spaces, which poses additional challenges to the analysis of SGDD. To address this challenge, we adopt a generalized definition of Fisher information [3, 17] and analyze the convergence of SGDD to the stationary distribution using a more general technique that encompasses both continuous and discrete settings.

We define a distribution $\mu_t$ over $\mathcal{X}$ that evolves according to likelihood steps and prior steps alternatively. We assume each likelihood step is implemented with the Metropolis-Hastings algorithm, and that each prior step is solved by the Euler method with an approximated score function. We compare $\mu_t$ to the continuous-time stationary distribution $\pi_t$, which alternates between $\pi^X$ and $\pi^Z$.

**Theorem 1.** *Consider running $K$ iterations of SGDD with a fixed $\eta > 0$ and an estimated concrete score $s_\theta(\mathbf{x}; t)$, and suppose that each prior step is solved by an $H$ step Euler method. Let $t^* > 0$ with $\sigma(t^*) = \eta$. Define $\pi_t$ and $\mu_t$ as stationary and non-stationary distributions. Over $K$ iterations of SGDD, the average relative Fisher information between $\mu_t$ and $\pi_t$ has*

$$\frac{1}{K} \sum_{k=0}^{K-1} \underbrace{\frac{1}{t^*} \int_{k(t^*+1)+1}^{(k+1)(t^*+1)} \mathsf{FI}_{\bar{Q}_t}(\pi_t \| \mu_t) \mathrm{d}t}_{\text{average Fisher divergence in the k-th prior step}} \leqslant \underbrace{\frac{2\mathsf{KL}(\pi_0\|\mu_0)}{Kt^*}}_{\text{convergence by } O(1/K)} + \underbrace{\frac{4M\epsilon}{c}}_{\text{score error}} + \underbrace{\frac{2MLt^*}{cH}}_{\text{discretization error}} . \quad (14)$$

*where $\left\| \frac{s_\theta(\cdot;t)-s(\cdot;t)}{s(\cdot;t)} \right\|_\infty \leqslant \epsilon \leqslant 1$, and $L, M, c$ are positive constants defined in Appendix A.*

We include the proof in Appendix A. Theorem 1 states that the average Fisher divergence of the non-stationary process to the stationary process in all prior steps converges at the rate of $O(1/K)$, up to a constant error term. This result extends our theoretical understanding of diffusion posterior sampling by generalizing the analysis [43, 51] on SDEs to general Markov processes using the free-energy-rate-functional-relative-Fisher-information (FIR) inequality [16]. Moreover, compared to existing analyses for continuous diffusion models, our analysis accounts not only for the imperfect score function but also for the discretization error in solving CTMCs as well.

Unlike some previous methods [27, 50] that rely on assumptions of surrogate value functions and an infinite number of particles, SGDD does not rely on such approximations and eventually converges to the posterior distribution with robustness to score approximation and discretization error. Consequently, SGDD can be viewed as a natural and principled approach for posterior sampling using discrete diffusion models, leading to empirical benefits as we demonstrate in Section 4.

## 4 Experiments

We conduct experiments to demonstrate the effectiveness of our algorithm on various posterior sampling tasks in discrete-state spaces, including discrete inverse problems and conditional generation guided by a reward function.

### 4.1 Experimental Setup

We evaluate our method across multiple domains, including synthetic data, DNA sequences, discrete image prior, and monophonic music, as listed in Table 1. We define the likelihood function for all inverse problems as $p(\mathbf{y}|\mathbf{x}) \propto \exp(-\|\boldsymbol{G}(\mathbf{x}) - \mathbf{y}\|/\sigma_\mathbf{y})$, while the likelihood functions for reward guiding tasks are $p(\mathbf{y}|\mathbf{x}) \propto \exp(\beta r(\mathbf{x}))$.

**Pretrained models.** For synthetic data, we use a closed-form concrete score function. For real datasets, we train a SEDD [29] model with the uniform transition kernel (details in Appendix C).

**Baselines.** We compare our methods to existing approaches for discrete diffusion posterior sampling: DPS [7], SVDD-PM [27], and SMC [50]. The original version of DPS relies on gradient back-propagation through both the likelihood function and the pretrained diffusion model, making it unsuitable for discrete-domain posterior sampling. In our experiments, we consider its discrete analogy by replacing the gradient with a guidance rate matrix. SVDD-PM is originally designed to sample from a reward-guided distribution $p^\beta(\mathbf{x}) \propto p(\mathbf{x}) \exp(\beta r(\mathbf{x}))$, but it is also adaptable to solving inverse problems. More implementation details are elaborated in Appendix C.2.

**SGDD implementation details.** We implement our method with a total of $K$ iterations. In each prior sampling step, we simulate the reverse continuous-time Markov chain with $H = 20$ steps. Additional details on hyperparameter choices are provided in Appendix C.3.

Table 1: A summary of posterior sampling tasks considered in our experiments.

| Domain | Synthetic | DNA enhancers | Images | Monophonic music |
|---|---|---|---|---|
| Task type | Inverse problem | Reward guidance | Inverse problem | Inverse problem |
| Dimensionality | $50^{10}$ | $4^{200}$ | $2^{1024}$ | $129^{256}$ |

Table 2: **Quantitative evaluation of posterior sampling on synthetic dataset** for sequence lengths of $D = 2, 5$ and 10. The *Hellinger distance* and *total variation* are computed between the empirical distribution of 10k samples and the true posterior distribution on the first two dimensions.

| | $D = 2$ | | $D = 5$ | | $D = 10$ | |
|---|---|---|---|---|---|---|
| | Hellinger ↓ | TV ↓ | Hellinger ↓ | TV ↓ | Hellinger ↓ | TV ↓ |
| SVDD-PM ($M = 20$) | 0.297 | 0.320 | 0.413 | 0.458 | 0.455 | 0.503 |
| SVDD-PM ($M = 100$) | 0.275 | 0.292 | 0.401 | 0.445 | 0.448 | 0.494 |
| SMC | 0.238 | 0.248 | 0.353 | 0.391 | 0.412 | 0.460 |
| DPS | 0.182 | 0.176 | 0.345 | 0.383 | 0.410 | 0.453 |
| SGDD | **0.149** | **0.125** | **0.214** | **0.222** | **0.334** | **0.365** |

## 4.2 Controlled Study using Synthetic Data

To evaluate the accuracy of SGDD in posterior sampling, we first conduct a case study on a synthetic task where the true posterior distribution is accessible. Let $\mathbf{x} \in \mathcal{X}^D$ follow a prior distribution $p(\mathbf{x})$, obtained by discretizing a Gaussian distribution $\mathcal{N}(0, \sigma^2 \boldsymbol{I}_D)$ over an $D$-dimensional grid. We define a forward model by $\boldsymbol{G}(\mathbf{x}) = \|f(\mathbf{x})\|_1$ where $f(\mathbf{x})$ maps $\mathbf{x}$ to $\mathbb{R}^D$.

We draw 10k independent samples with each algorithm and compute the frequency map for the first two dimensions of $\mathbf{x}$. We measure the *Hellinger distance* and *total variation distance* between the empirical distribution and the ground truth posterior distribution. As shown in Table 2, our method achieves substantially more accurate posterior sampling compared to baseline methods. We include visualization of the generated samples in Appendix D.3 (Fig. 8) by projecting samples onto the first two dimensions and computing their heatmaps. When the number of dimensions is small, e.g., $D = 2$, all methods approximate the true posterior distribution well. However, as the dimensionality increases, DPS, SMC, and SVDD-PM struggle to capture the posterior and tend to degenerate to the prior distribution. While SGDD consistently outperforms baseline methods in terms of *Hellinger distance* and *total variation distance*, it also shows a slight deviation from the posterior distribution as the dimensionality increases.

## 4.3 Guided DNAs Generation

We evaluate our method for generating regulatory DNA sequences that drive gene expression in specific cell types, which is an important task for cell and gene therapy and commonly studied in the literature [27, 44, 48].

We train a discrete diffusion model [29] on a publicly available dataset from Gosai et al. [15] that consists of ∼700k DNA sequences. The expression driven by each sequence in human cell lines is measured by massively parallel reporter assays (MPRAs). Following Wang et al. [48], we split the dataset into two subsets and train surrogate reward oracles to predict the activity level in the HepG2 cell line. We use the oracle trained on the training set, $r(\cdot)$, to guide our diffusion model, and evaluate its performance using the other one trained on the test dataset. The likelihood function is therefore defined as $p(\mathbf{y}|\mathbf{x}) \propto \exp(\beta r(\mathbf{x}))$.

**Evaluation metrics.** We evaluate the quality of generated samples using the following metrics:

- *Predicted activity (Pred-Activity).* We evaluate the activity level of generated sequences using the oracle trained on the test dataset. This corresponds to the reward function that we aim to optimize.
- *Binary classification on chromatin accessibility (ATAC-Acc).* Following [48], we use a binary classifier trained on chromatin accessibility data in the HepG2 cell line. As active enhancers should have accessible chromatin, *ATAC-Acc* can be used to validate the generated sequences.
- *Log-likelihood.* We estimate the log-likelihood of the generated samples using the ELBO of a pretrained masked diffusion model [48]. This regulates the generated samples to stay close to the prior distribution of natural DNA sequences.

We include more baseline methods that require additional training for this task, including a discrete diffusion fine-tuning method, DRAKES [48], classifier guidance (CG) for discrete diffusion models [32], and twisted diffusion sampler (TDS) [50] which uses CG as the proposal function for sequential Monte Carlo sampling. We further include a Feynman-Kac diffusion sampler [33] with MAX potential function as well.

**Results.** We generate 640 DNA samples using different algorithms and compute the median and average *Pred-Activity* of the generated sequences. As shown in Table 3, our method achieves the

Table 3: **Quantitative results for guided DNA (enhancers) sampling.** We generate 640 independent samples with different algorithms according to the reward-shaped posterior distribution. We report the mean and median of the enhancer's activity computed by the surrogate reward.

| | Pred-Activity (median) ↑ | Pred-Activity (average) ↑ | ATAC Acc % ↑ | Log-likelihood (average) ↑ |
|---|---|---|---|---|
| SVDD-PM | 5.41 | $5.08_{+1.51}$ | 49.9 | $\mathbf{-241}_{+47}$ |
| SMC | 4.15 | $3.99_{+1.49}$ | 39.9 | $-259_{+13}$ |
| CG | 2.90 | $2.76_{+0.94}$ | 0.0 | $-265_{+5}$ |
| TDS | 4.64 | $4.62_{+1.09}$ | 45.3 | $-257_{+9}$ |
| FK-MAX | 4.90 | $4.79_{+1.02}$ | 47.1 | $-243_{+9}$ |
| DRAKES w/o KL | 6.44 | $6.28_{+0.76}$ | 82.5 | $-281_{+6}$ |
| DRAKES w/ KL | 5.61 | $5.24_{+1.39}$ | 92.5 | $-264_{+9}$ |
| SGDD ($\beta = 30$) | 8.82 | $8.69_{+1.12}$ | 91.7 | $-\underline{2}46_{+41}$ |
| SGDD ($\beta = 50$) | **9.14** | $\mathbf{8.96}_{+1.17}$ | **93.0** | $-261_{+29}$ |

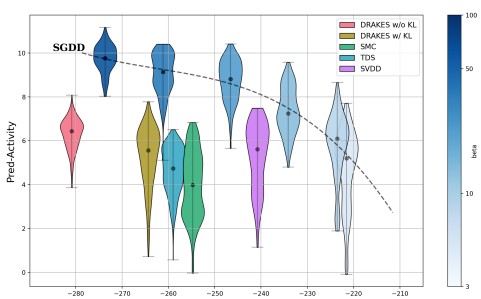

Figure 2: **Generating DNA sequences involves trade-offs** between targeted objectives and consistency with the prior distribution. SGDD is plotted in blue and darker colors mean higher $\beta$.

Figure 3: **Diversified samples when the measurement y is sparse.** Samples are generated by SGDD when solving an MNIST inpainting task.

highest *Pred-Activity* (reward values) compared to existing baselines, including 42% higher median reward compared to the previous state-of-the-art method.

Moreover, as SGDD samples from the posterior distribution, $p(\mathbf{x}) \exp(\beta r(\mathbf{x}))/Z$, the guidance strength $\beta$ controls the strength of guidance imposed in the sampling algorithm. We instantiate SGDD with $\beta$ ranging from 3 to 100 and calculate the *Pred-Activity* and *Log-likelihood* of the generated samples. We plot the distribution of the samples' *Pred-Activity* with various $\beta$ in Fig. 2. In general, using a larger $\beta$ results in higher *Pred-Activity* (reward), at the expense of a lower likelihood as the target distribution deviates further from the prior distribution.

### 4.4 Discrete Image Inverse Problems

**Setup.** We evaluate on a discretized image domain. Specifically, we convert the MNIST dataset [25] to binary strings by discretizing the images, and train a discrete diffusion prior on 60k training data. As examples of linear and nonlinear forward models on binary strings, we consider AND and XOR operators. We randomly pick $\gamma D$ pairs of positions $(i_p, j_p)$ over $\{1, \ldots, D\}$, and compute:

$$\boldsymbol{G}_{\text{AND}}(\mathbf{x}) = [\mathbf{x}_{i_p} \wedge \mathbf{x}_{j_p}]_{p=1,\ldots,\gamma D}, \quad \boldsymbol{G}_{\text{XOR}}(\mathbf{x}) = [\mathbf{x}_{i_p} \oplus \mathbf{x}_{j_p}]_{p=1,\ldots,\gamma D}. \tag{15}$$

These logical operators are challenging to approximate using surrogate functions to guide sampling.

**Evaluation metrics.** We use 1k binary images from the test set of MNIST and calculate the *peak signal-to-noise ratio (PSNR)* of the reconstructed image. Furthermore, we train a simple convolutional neural network on MNIST as a surrogate and report the *classifier accuracy* of the generated samples.

**Results.** As shown in Table 4, SGDD outperforms baseline methods by a large margin in both XOR and AND tasks. We present the samples generated by SGDD for the XOR task in Appendix D.2. The reconstructed samples are visually consistent with the underlying ground truth signal, which explains the high class accuracy of our method shown in Table 4.

**Sample diversity.** Furthermore, we demonstrate that SGDD generates diversified samples when the measurement y is sparse. For example, when a digit is masked with a large box, as shown

Table 4: **Quantitative results for XOR and AND problems on discretized MNIST.** We report the mean and standard deviation of *PSNR* and *class accuracy* across 1k generated samples. SGDD demonstrates superior performance on both tasks.

| | XOR | | AND | |
| --- | --- | --- | --- | --- |
| | PSNR ↑ | Accuracy (%) ↑ | PSNR ↑ | Accuracy (%) ↑ |
| SVDD-PM ($M = 20$) | $11.81_{\pm 2.54}$ | 51.4 | $10.04_{\pm 1.49}$ | 33.7 |
| SMC | $10.05_{\pm 1.54}$ | 27.8 | $10.25_{\pm 1.63}$ | 24.4 |
| DPS | $9.04_{\pm 1.21}$ | 30.0 | $8.67_{\pm 0.91}$ | 24.5 |
| SGDD | $\mathbf{20.17}_{\pm 3.47}$ | **91.2** | $\mathbf{17.25}_{\pm 3.82}$ | **79.4** |

Table 5: **Quantitative evaluation of infilling monophonic music sequences.**

| | $\gamma = 40\%$ | | $\gamma = 60\%$ | |
| --- | --- | --- | --- | --- |
| | Hellinger ↓ | Outliers ↓ | Hellinger ↓ | Outliers ↓ |
| SVDD-PM | $0.465_{\pm 0.129}$ | $0.112_{\pm 0.098}$ | $0.529_{\pm 0.137}$ | $0.114_{\pm 0.119}$ |
| SMC | $0.492_{\pm 0.143}$ | $0.152_{\pm 0.138}$ | $0.552_{\pm 0.131}$ | $0.164_{\pm 0.140}$ |
| SGDD | $\mathbf{0.126}_{\pm 0.098}$ | $\mathbf{0.003}_{\pm 0.002}$ | $\mathbf{0.244}_{\pm 0.164}$ | $\mathbf{0.060}_{\pm 0.138}$ |

in Fig. 3, the measurement lacks sufficient information to fully recover the original digit. In this scenario, SGDD generate samples from multiple plausible modes, including digits 1,4,7 and 9. This highlights the ability of our method to produce diverse samples from the posterior distribution while preserving consistency with the measurement.

**Higher dimensional images.** To demonstrate the scalability of SGDD, we further evaluate SGDD on a higher-dimensional dataset, FFHQ 256 [21] in Appendix D.1, where we solve the super-resolution task by operating in the discrete latent space of a pretrained vector-quantized diffusion model.

### 4.5 Monophonic Music

We conduct experiments on monophonic songs from the Lakh pianoroll dataset [11]. Following the preprocessing steps in [4], we obtain monophonic musical sequences of length $D = 256$ and vocabulary size $N = 129$. The ordering of musical notes is scrambled to destroy any ordinal structure. Therefore, the likelihood function for this task is $p(\mathbf{y}|\mathbf{x}) \propto \exp(-\|\boldsymbol{G}(\mathbf{x}) - \mathbf{y}\|_0/\sigma_{\mathbf{y}})$. This $\ell_0$ norm makes this inverse problem ill-posed, in stark contrast to most reward-guided generation tasks, where the reward functions are much smoother. We evaluate our method on an inpainting task, where the forward model $\boldsymbol{G}(\cdot)$ randomly masks $\gamma = 40\%, 60\%$ of the notes. As in Campbell et al. [4], we use the *Hellinger distance of histograms* and the *proportion of outlier notes* as metrics to evaluate our method. We run experiments on 100 samples in the test set and report the quantitative results in Table 5. SGDD successfully completes monophonic music sequences in a style more consistent with the given conditions, achieving a 2x reduction in Hellinger distance compared to SVDD-PM.

### 4.6 More Ablation Studies

We conduct more ablation studies on our method to investigate how specific design choices may affect its performance. We discuss the effectiveness of using an annealing noise schedule $\eta_k$ over a fixed noise $\eta_k \equiv \eta$ in Appendix B.1, including empirical discovery and theoretical justification. We include runtime analysis of SGDD in Appendix B.2, showing the tradeoff between sample quality and computing budgets regarding NFEs. Finally, we highlight the value of using an informative discrete diffusion prior in solving inverse problems by comparing SGDD to MCMC methods without a prior (Appendix B.3).

## 5 Conclusion

We introduce SGDD, a principled discrete diffusion posterior sampling algorithm based on the split Gibbs sampler, extending plug-and-play diffusion methods from Euclidean spaces to discrete-state spaces. Our algorithm alternates between prior and likelihood sampling steps with decreasing regularization level $\eta_k$. By carefully designing the regularization potential function, we establish a connection between prior sampling steps and discrete diffusion samplers, integrating discrete diffusion models into the split Gibbs sampling framework. We prove the convergence of SGDD to the targeted distribution. We evaluate our method on diverse tasks from various domains, including solving inverse problems and sequence generation guided by reward functions, where it significantly outperforms existing baselines.

## Acknowledgments

This research was supported in part by gifts from OpenAI, Two Sigma, and Cisco. Z.W. is supported by the Amazon AI4Science fellowship. W.C. is supported by the Kortschak Scholars Fellowship.

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

# A Theory

**Notations.** We consider a linear stochastic process whose forward Kolmogorov equation can be written as $\partial_t \pi_t = \boldsymbol{Q}_t \pi_t$ with boundary condition $\pi_{t=0} = \pi_0$, where $\pi_t \sim \mathcal{P}(\mathcal{X})$. For the simplicity of notation, we let $\boldsymbol{Q}_t$ to generate *reverse continuous-time Markov chain*, which means $\boldsymbol{Q}_t^{[ij]} = s(\mathbf{x}_j; T-t)_{\mathbf{x}_i} \boldsymbol{Q}_{\text{uniform}}^{[ji]}$; and time $t$ flows with the sampling algorithm, which means $\pi_0$ is a uniform distribution and $\pi_T$ is the data distribution.

For two probability mass functions $\mu$ and $\pi$, we discuss their KL divergence,

$$\mathsf{KL}(\mu \| \pi) := \mathbb{E}_{\mathbf{x} \sim \mu} \left[ \log \frac{\mu}{\pi}(\mathbf{x}) \right]. \tag{16}$$

We define the *relative Fisher information* between $\mu$ and $\pi$ as

$$\mathsf{FI}_{\boldsymbol{Q}}(\mu \| \pi) := \sum_{\mathbf{x}_i, \mathbf{x}_j \in \mathcal{X}} \pi(\mathbf{x}_i) \boldsymbol{Q}^{[j,i]} \left( f(\mathbf{x}_j) - f(\mathbf{x}_i) - f(\mathbf{x}_i) \log \frac{f(\mathbf{x}_j)}{f(\mathbf{x}_i)} \right), \; f := \mu/\pi. \tag{17}$$

Note that when $\boldsymbol{Q}$ is irreducible, the relative Fisher information $\mathsf{FI}_{\boldsymbol{Q}}(\mu \| \pi) \geqslant 0$, and $\mathsf{FI}_{\boldsymbol{Q}}(\mu \| \pi) = 0$ if and only if $\mu = \pi$.

In continuous state spaces, the relative Fisher information criterion has been used to derive general first-order guarantees for non-log-concave sampling [2]. This line of analysis has been adapted to provide theoretical insights for posterior sampling methods using diffusion models [43, 51]. The formula in Eq. (17) represents a discrete state space analogue of Fisher information. We refer interested readers to [3, 10, 17] for more discussions on this topic and the relation between KL and Fisher information in the discrete state space. Our analysis in this paper adapts these techniques and provides general first-order guarantees for posterior sampling with discrete diffusion models.

**Time interpolation of SGDD.** SGDD alternates between likelihood steps and prior steps. Let $t^* > 0$ such that $\sigma_{t*} = \eta$. We define $\{\pi_t\}$ as the distributions at time $t$ of the stationary process, and $\{\mu_t\}$ as the distributions of the non-stationary process.

- In time intervals $\tau \in [k(t^*+1)+1, (k+1)(t^*+1)]$. The stationary distribution is initialized with $\pi_{k(t*+1)+1}(\mathbf{x}) = \pi_\eta^Z(\mathbf{x})$. We run a prior step on $\mu_\tau$ with the learned concrete score function for $H$ steps, while $\pi_\tau$ evolves in continuous time with the true concrete score function.
- In time intervals $\tau \in [k(t^*+1), k(t^*+1)+1]$, we run a Metropolis-Hastings sampling algorithm on both $\pi_\tau$ and $\mu_\tau$.

**Assumptions.** Our analysis relies on the following assumptions:

(i) Concrete score is well estimated: $\left\| \frac{s_\theta(\cdot;t) - s(\cdot;t)}{s(\cdot;t)} \right\|_\infty \leqslant \epsilon < 1$.

(ii) Smoothness of score function in $t$: $\| \frac{s(\cdot;t+\Delta t) - s(\cdot;t)}{s(\cdot;t)} \|_\infty \leqslant L \cdot \Delta t$.

(iii) Strong irreducibility: $\boldsymbol{Q}_t^{[i,j]} > 0$ for $i \neq j$.

(iv) Bounded probability ratio: $\sup_t \| \log \frac{\mu_t(\mathbf{x})}{\pi_t(\mathbf{x})} \|_\infty \leqslant B$.

(v) The entry-wise absolute of the reverse-time transition matrix is bounded: $\sup_t \||\boldsymbol{Q}_t\||_1 \leqslant M$.

## A.1 Lemmas

**Lemma 1** (Data processing inequality of Metropolis Hasting). *Running Metropolis Hasting on two distributions $\pi_\tau$ and $\mu_\tau$ does not increase their KL divergence, i.e.,*

$$\mathsf{KL}(\pi_{k(t*+1)} \| \mu_{k(t*+1)}) \geqslant \mathsf{KL}(\pi_{k(t*+1)+1} \| \mu_{k(t*+1)+1}). \tag{18}$$

**Lemma 2** (Free-energy-rate-functional-relative-Fisher-information (FIR) inequality (from Theorem 6.2.3. in [16])). *Consider two continuous time Markov chains: $\partial_t \pi_t = \boldsymbol{Q}_t \pi_t$ and $\partial_t \mu_t = \tilde{\boldsymbol{Q}}_t \mu_t$. Suppose Assumption (iv) holds, then there exists a constant $c > 0$, such that*

$$\partial_t \mathsf{KL}(\pi_t \| \mu_t) \leqslant -\frac{1}{2} \mathsf{FI}_{\tilde{\boldsymbol{Q}}_t}(\pi_t \| \mu_t) + \frac{2}{c} \mathcal{L}_{\tilde{\boldsymbol{Q}}_t}(\pi_t, \boldsymbol{Q}_t \pi_t), \tag{19}$$

*where $\mathcal{L}_{\tilde{\mathbf{Q}}}(\pi_t, \partial_t \pi_t) \geqslant 0$ is the Lagrangian defined as*

$$\mathcal{L}_{\tilde{\mathbf{Q}}}(\pi_t, \partial_t \pi_t) = \sup_{\varphi \in \mathcal{C}_b(\mathcal{X})} \left[ \langle \varphi, \partial_t \pi_t \rangle - \sum_{\mathbf{x}_i, \mathbf{x}_j \in \mathcal{X}} \pi_t(\mathbf{x}_i) \tilde{\mathbf{Q}}^{[j,i]} \exp(\varphi(\mathbf{x}_j) - \varphi(\mathbf{x}_i)) \right]. \tag{20}$$

*Proof Sketch.* This lemma follows from Theorem 6.2.3. in Hilder et al. [16]. For completeness, we provide a sketch of the proof here.

By a direct calculation of derivatives, we have

$$\partial_t \mathsf{KL}(\pi_t \| \mu_t) = \partial_t \sum_{\mathbf{x}_i} \pi_t(\mathbf{x}_i) \log \frac{\pi_t(\mathbf{x}_i)}{\mu_t(\mathbf{x}_i)} = \sum_{\mathbf{x}_i} \left( \partial_t \pi_t(\mathbf{x}_i) \log \frac{\pi_t(\mathbf{x}_i)}{\mu_t(\mathbf{x}_i)} - \frac{\pi_t(\mathbf{x}_i)}{\mu_t(\mathbf{x}_i)} \partial_t \mu_t(\mathbf{x}_i) \right)$$

$$= \sum_{\mathbf{x}_i, \mathbf{x}_j} \left( \mathbf{Q}_t^{[i,j]} \pi_t(\mathbf{x}_j) \log \frac{\pi_t(\mathbf{x}_i)}{\mu_t(\mathbf{x}_i)} - \frac{\pi_t(\mathbf{x}_i)}{\mu_t(\mathbf{x}_i)} \tilde{\mathbf{Q}}_t^{[i,j]} \mu_t(\mathbf{x}_j) \right)$$

$$= \sum_{\mathbf{x}_i, \mathbf{x}_j} \tilde{\mathbf{Q}}_t^{[i,j]} \mu_t(\mathbf{x}_j) \left( \frac{\pi_t(\mathbf{x}_i)}{\mu_t(\mathbf{x}_i)} \log \frac{\pi_t(\mathbf{x}_i)}{\mu_t(\mathbf{x}_i)} - \frac{\pi_t(\mathbf{x}_i)}{\mu_t(\mathbf{x}_i)} \right) - \sum_{\mathbf{x}_i, \mathbf{x}_j} (\tilde{\mathbf{Q}}_t^{[i,j]} - \mathbf{Q}_t^{[i,j]}) \pi_t(\mathbf{x}_j) \log \frac{\pi_t(\mathbf{x}_i)}{\mu_t(\mathbf{x}_i)}$$

Using the fact that $\sum_{\mathbf{x}_i} \tilde{\mathbf{Q}}_t^{[i,j]} = 0$ and the definition in Eq. (17), we have the equality

$$\sum_{\mathbf{x}_i, \mathbf{x}_j} \tilde{\mathbf{Q}}_t^{[i,j]} \mu_t(\mathbf{x}_j) \left( \frac{\pi_t(\mathbf{x}_i)}{\mu_t(\mathbf{x}_i)} \log \frac{\pi_t(\mathbf{x}_i)}{\mu_t(\mathbf{x}_i)} - \frac{\pi_t(\mathbf{x}_i)}{\mu_t(\mathbf{x}_i)} \right) = -\mathsf{FI}_{\tilde{\mathbf{Q}}_t}(\pi_t \| \mu_t)$$

Using the relation $\partial_t \pi_t = \mathbf{Q}_t \pi_t$ this leads to

$$\partial_t \mathsf{KL}(\pi_t \| \mu_t) = -\mathsf{FI}_{\tilde{\mathbf{Q}}_t}(\pi_t \| \mu_t) - \underbrace{(\log \frac{\pi_t}{\mu_t})^T (\tilde{\mathbf{Q}}_t - \partial_t) \pi_t}_{\text{error term}}. \tag{21}$$

This formula is similar to Lemma 1 in Yau [53].

We define

$$\mathcal{L}_{\tilde{\mathbf{Q}}_t}(\pi_t, \partial_t \pi_t) = \sup_{\varphi \in \mathcal{C}_b(\mathcal{X})} [\langle \varphi, \partial_t \pi_t \rangle - (e^{-\varphi} \pi_t)^T \tilde{\mathbf{Q}}_t^T e^{\varphi}]$$

$$= \sup_{\varphi \in \mathcal{C}_b(\mathcal{X})} [\langle \varphi, \partial_t \pi_t - \tilde{\mathbf{Q}}_t \pi_t \rangle - \underbrace{\pi_t^T (e^{-\varphi} \tilde{\mathbf{Q}}_t^T e^{\varphi} - \tilde{\mathbf{Q}}_t^T \varphi)}_{\text{denoted as } \tilde{\mathcal{H}}(\pi_t, \varphi)}] \tag{22}$$

By the variational characterization of the Lagrangian, for any continuous, bounded $\varphi$,

$$\langle \varphi, \partial_t \pi_t - \tilde{\mathbf{Q}}_t \pi_t \rangle \leqslant \mathcal{L}_{\tilde{\mathbf{Q}}_t}(\pi_t, \partial_t \pi_t) + \tilde{\mathcal{H}}(\pi_t, \varphi). \tag{23}$$

If we choose $\varphi = \log \frac{\pi_t}{\mu_t}$, the inequality above gives a bound to the error term in Eq. (21). Moreover, it is easy to verify that

$$\tilde{\mathcal{H}}(\pi_t, \log \frac{\pi_t}{\mu_t}) = \mathsf{FI}_{\tilde{\mathbf{Q}}_t}(\pi_t \| \mu_t). \tag{24}$$

In fact, when the space $\mathcal{X}$ is continuous, choosing $\varphi(\mathbf{x}) = \log \frac{\pi_t}{\mu_t}(\mathbf{x})$ exactly recovers Lemma A.4 in [51]. However, as pointed out in [16], it is necessary to consider a rescaled $\varphi = \lambda \log \frac{\pi_t}{\mu_t}$ with $\lambda \in (0, 1)$ to derive a bound in finite space $\mathcal{X}$.

Fortunately, according to Lemma 6.2.2. in [16], for any $\varphi \in \mathcal{C}_b(\mathcal{X})$ with $\|\varphi\|_\infty \leqslant B$, there exists a positive constant $c = c(B) \in (0, 1)$, such that

$$\tilde{\mathcal{H}}(\pi_t, \lambda\varphi) \leqslant \frac{\lambda^2}{c} \tilde{\mathcal{H}}(\pi_t, \varphi). \tag{25}$$

Plugging in $\varphi = \lambda \log \frac{\pi_t}{\mu_t}$ in Eq. (23), we have

$$\langle \lambda \log \frac{\pi_t}{\mu_t}, \partial_t \pi_t - \tilde{\mathbf{Q}}_t \pi_t \rangle \leqslant \mathcal{L}_{\tilde{\mathbf{Q}}_t}(\pi_t, \partial_t \pi_t) + \tilde{\mathcal{H}}(\pi_t, \lambda \log \frac{\pi_t}{\mu_t}) \tag{26}$$

$$\leqslant \mathcal{L}_{\tilde{\mathbf{Q}}_t}(\pi_t, \partial_t \pi_t) + \frac{\lambda^2}{c} \tilde{\mathcal{H}}(\pi_t, \log \frac{\pi_t}{\mu_t}) \tag{27}$$

Combine this with Eq. (21),

$$\partial_t \mathsf{KL}(\pi_t \| \mu_t) \leqslant -\mathsf{FI}_{\tilde{\boldsymbol{Q}}_t}(\pi_t \| \mu_t) + \frac{1}{\lambda}\mathcal{L}_{\tilde{\boldsymbol{Q}}_t}(\pi_t, \partial_t \pi_t) + \frac{\lambda}{c}\tilde{\mathcal{H}}(\pi_t, \log \frac{\pi_t}{\mu_t})$$

$$= -(1 - \frac{\lambda}{c})\mathsf{FI}_{\tilde{\boldsymbol{Q}}_t}(\pi_t \| \mu_t) + \frac{1}{\lambda}\mathcal{L}_{\tilde{\boldsymbol{Q}}_t}(\pi_t, \partial_t \pi_t). \tag{28}$$

Finally, choosing $\lambda = \frac{c}{2} \in (0, 1)$ recovers the statement of Lemma 2.

$\square$

**Lemma 3.** *When the learned score function $s_\theta$ satisfies Assumption (i) and (v),*

$$\mathcal{L}_{\boldsymbol{Q}_t}(\mu_t, \tilde{\boldsymbol{Q}}_t \mu_t) \leqslant M\epsilon. \tag{29}$$

*Proof.* By definition,

$$\mathcal{L}_{\boldsymbol{Q}_t}(\mu_t, \tilde{\boldsymbol{Q}}_t \mu_t) = \sup_{\varphi \in \mathcal{C}_b(\mathcal{X})} \sum_{\mathbf{x}_i, \mathbf{x}_j \in \mathcal{X}} \left[ \mu_t(\mathbf{x}_i)\tilde{\boldsymbol{Q}}_t^{[j,i]}\varphi(\mathbf{x}_j) - \mu_t(\mathbf{x}_i)\boldsymbol{Q}_t^{[j,i]}e^{\varphi(\mathbf{x}_j)-\varphi(\mathbf{x}_i)} \right]$$

$$= \sup_{\varphi \in \mathcal{C}_b(\mathcal{X})} \sum_{\mathbf{x}_i, \mathbf{x}_j \in \mathcal{X}} \left[ \mu_t(\mathbf{x}_i)\tilde{\boldsymbol{Q}}_t^{[j,i]}(\varphi(\mathbf{x}_j) - \varphi(\mathbf{x}_i)) - \mu_t(\mathbf{x}_i)\boldsymbol{Q}_t^{[j,i]}e^{\varphi(\mathbf{x}_j)-\varphi(\mathbf{x}_i)} \right]$$

$$(\mathbf{1}^T \tilde{\boldsymbol{Q}}_t \mathbf{u} = 0 \text{ for any } \mathbf{u})$$

$$= \sup_{\varphi \in \mathcal{C}_b(\mathcal{X})} \sum_{\mathbf{x}_i \neq \mathbf{x}_j} \left[ \mu_t(\mathbf{x}_i)\boldsymbol{Q}_t^{[j,i]}\left( -\frac{\tilde{\boldsymbol{Q}}_t^{[j,i]}}{\boldsymbol{Q}_t^{[j,i]}}z_{ij} - e^{-z_{ij}} \right) \right] + \sum_{\mathbf{x}_i} \mu_t(\mathbf{x}_i)(\tilde{\boldsymbol{Q}}_t^{[i,i]} - \boldsymbol{Q}_t^{[i,i]})$$

$$(z_{ij} = \varphi(\mathbf{x}_i) - \varphi(\mathbf{x}_j))$$

$$\leqslant \sum_{\mathbf{x}_i \neq \mathbf{x}_j} \left[ \mu_t(\mathbf{x}_i)\boldsymbol{Q}_t^{[j,i]} \cdot \sup_{z_{ij} \in \mathbb{R}} \left( -\frac{\tilde{\boldsymbol{Q}}_t^{[j,i]}}{\boldsymbol{Q}_t^{[j,i]}}z_{ij} - e^{-z_{ij}} \right) \right] + \epsilon \sum_{\mathbf{x}_i} \mu_t(\mathbf{x}_i)|\boldsymbol{Q}_t^{[i,i]}|. \tag{30}$$

where the inequality is due to swapping sup and summation, and that $\mu_t(\mathbf{x}_i)\boldsymbol{Q}_t^{[j,i]} \geqslant 0$ for $i \neq j$.

Consider the function $g(z) = -uz - e^{-z}$. When $u \geqslant 0$, this function is maximized when $z = -\log u$. Using $u = \boldsymbol{Q}_t^{[j,i]}/\tilde{\boldsymbol{Q}}_t^{[j,i]}$,

$$\sup_{z_{ij}} \left( -\frac{\boldsymbol{Q}_t^{[j,i]}}{\tilde{\boldsymbol{Q}}_t^{[j,i]}}z_{ij} - e^{-z_{ij}} \right) = (u-1)\log u \leqslant |u-1| \leqslant \epsilon, \tag{31}$$

for $\epsilon \leqslant 1$. Combining with Eq. (30), we have

$$\mathcal{L}_{\boldsymbol{Q}_t}(\mu_t, \tilde{\boldsymbol{Q}}_t \mu_t) \leqslant \epsilon \mathbf{1}^T |\boldsymbol{Q}_t| \mu_t \leqslant \epsilon \|\boldsymbol{Q}_t\|_1 \|\mu_t\|_1 \leqslant \epsilon M. \tag{32}$$

$\square$

### Additional Notes on Lemma 3.

With a slightly different derivation, assumptions (i) and (v) can potentially be replaced by the following assumption that links directly to the score entropy loss defined in Definition 3.1 of Lou et al. [29].

### Alternative assumption.

(i′) Score function is well estimated, i.e., the score entropy loss for distributions on the stationary process is bounded:

$$\mathbb{E}_{\mathbf{x}_i \sim \pi_t}\mathsf{SE}_{\boldsymbol{Q}^{\mathsf{fw}}}(\mathbf{x}_i; \theta, t) := \mathbb{E}_{\mathbf{x}_i \sim \pi_t} \sum_{\mathbf{x}_j \neq \mathbf{x}_i} K\left( \frac{s_\theta(\mathbf{x}_i; t)_{\mathbf{x}_j}}{s(\mathbf{x}_i; t)_{\mathbf{x}_j}} \right) \boldsymbol{Q}_t^{\mathsf{fw}\,[i,j]}s(\mathbf{x}_i; t)_{\mathbf{x}_j} \leqslant \epsilon_{\mathsf{SE}},$$

for any $t \in [0, 1]$, where $K(a) = a - 1 - \log a$.

**Lemma 4.** *When the learned score function $s_\theta$ satisfies Assumption (i′),*

$$\mathcal{L}_{\tilde{\boldsymbol{Q}}_t}(\pi_t, \boldsymbol{Q}_t \pi_t) \leqslant \epsilon_{\mathsf{SE}}. \tag{33}$$

*Proof.*

$$\mathcal{L}_{\tilde{\boldsymbol{Q}}_t}(\pi_t, \boldsymbol{Q}_t\pi_t) = \sup_{\varphi \in \mathcal{C}_b(\mathcal{X})} \sum_{\mathbf{x}_i \neq \mathbf{x}_j} \left[ \pi_t(\mathbf{x}_i)\tilde{\boldsymbol{Q}}_t^{[j,i]} \left( \frac{\boldsymbol{Q}_t^{[j,i]}}{\tilde{\boldsymbol{Q}}_t^{[j,i]}}(1 - z_{ij}) - e^{-z_{ij}} \right) \right] + \sum_{\mathbf{x}_i} \pi_t(\mathbf{x}_i)(\boldsymbol{Q}_t^{[i,i]} - \tilde{\boldsymbol{Q}}_t^{[i,i]})$$

$$\leqslant \sum_{\mathbf{x}_i \neq \mathbf{x}_j} \left[ \pi_t(\mathbf{x}_i)\tilde{\boldsymbol{Q}}_t^{[j,i]} \frac{s(\mathbf{x}_i;t)}{s_\theta(\mathbf{x}_i;t)} \log \frac{s(\mathbf{x}_i;t)}{s_\theta(\mathbf{x}_i;t)} \right] - \sum_{\mathbf{x}_i} \pi_t(\mathbf{x}_i) \sum_{j \neq i} (\boldsymbol{Q}_t^{[j,i]} - \tilde{\boldsymbol{Q}}_t^{[j,i]})$$

$$\tag{34}$$

$$= \sum_{\mathbf{x}_i \neq \mathbf{x}_j} \left[ \pi_t(\mathbf{x}_i)\boldsymbol{Q}_t^{[j,i]} \log \frac{s(\mathbf{x}_i;t)}{s_\theta(\mathbf{x}_i;t)} \right] - \sum_{\mathbf{x}_i} \pi_t(\mathbf{x}_i) \sum_{j \neq i} (\boldsymbol{Q}_t^{[j,i]} - \tilde{\boldsymbol{Q}}_t^{[j,i]})$$

$$= \sum_{\mathbf{x}_i \neq \mathbf{x}_j} \left[ \pi_t(\mathbf{x}_i)\boldsymbol{Q}_t^{\mathsf{fw}[i,j]} \left( s_\theta(\mathbf{x}_i;t)_{\mathbf{x}_j} \log \frac{s(\mathbf{x}_i;t)}{s_\theta(\mathbf{x}_i;t)} - s(\mathbf{x}_i;t) + s_\theta(\mathbf{x}_i;t) \right) \right]$$

$$= \mathbb{E}_{\mathbf{x}_i \sim \pi_t} \sum_{\mathbf{x}_j \neq \mathbf{x}_i} \left[ K\left( \frac{s_\theta(\mathbf{x}_i;t)}{s(\mathbf{x}_i;t)} \right) \boldsymbol{Q}_t^{\mathsf{fw}[i,j]} s(\mathbf{x}_i;t)_{\mathbf{x}_j} \right] \leqslant \epsilon_{\mathsf{SE}}. \tag{35}$$

where Eq. (34) comes from the previous Lemma 3. $\qquad\square$

## A.2 Proof of Theorem 1

*Proof.* Consider discretizing a prior step $[0, t^*]$ uniformly into $H$ intervals, $[h\delta, (h+1)\delta]$, for $h = 0, \dots, H-1$, where $\delta = t^*/H$. In the $h$-th interval,

$$\partial_t \mu_t = \tilde{\boldsymbol{Q}}_{h\delta}\mu_t. \tag{36}$$

Applying Lemma 2 on $\mu_t$ and $\pi_t$ in the $h$-th interval,

$$\partial_t \mathsf{KL}(\pi_t\|\mu_t) \leqslant -\frac{1}{2}\mathsf{FI}_{\tilde{\boldsymbol{Q}}_{h\delta}}(\pi_t\|\mu_t) + \frac{2}{c}\mathbb{E}_{\mathbf{x}_i \sim \pi_t}\mathcal{L}_{\tilde{\boldsymbol{Q}}_{h\delta}}(\pi_t, \boldsymbol{Q}_t\pi_t). \tag{37}$$

After integrating both sides by $t$ and rearranging, (for simplicity of notation, we denote $\tau_k = k(t^*+1)+1$ the starting time of the $k$-th prior step)

$$\int_{\tau_k+h\delta}^{\tau_k+(h+1)\delta} \mathsf{FI}_{\boldsymbol{Q}_t}(\mu_t\|\pi_t)\mathrm{d}t \leqslant \int_{k(t^*+1)+1+h\delta}^{k(t^*+1)+1+(h+1)\delta} \left[ -2\partial_t\mathsf{KL}(\mu_t\|\pi_t) + \frac{4}{c}\mathcal{L}_{\boldsymbol{Q}_t}(\mu_t, \tilde{\boldsymbol{Q}}_{\tau_k+h\delta}\mu_t) \right] \mathrm{d}t$$

$$= 2\Big(\mathsf{KL}(\mu_{\tau_k+h\delta}\|\pi_{\tau_k+h\delta}) - \mathsf{KL}(\mu_{\tau_k+(h+1)\delta}\|\pi_{\tau_k+(h+1)\delta})\Big) + \frac{4}{c}\int_{\tau_k+h\delta}^{\tau_k+(h+1)\delta} \mathcal{L}_{\boldsymbol{Q}_t}(\mu_t, \tilde{\boldsymbol{Q}}_{\tau_k+h\delta}\mu_t)\mathrm{d}t.$$

$$\tag{38}$$

Taking a summation over $h$,

$$\int_{k(t^*+1)+1}^{(k+1)(t^*+1)} \mathsf{FI}_{\boldsymbol{Q}_t}(\mu_t\|\pi_t) \leqslant 2\Big(\mathsf{KL}(\mu_{k(t^*+1)+1}\|\pi_{k(t^*+1)+1}) - \mathsf{KL}(\mu_{(k+1)(t^*+1)}\|\pi_{(k+1)(t^*+1)})\Big)$$

$$+ \sum_{h=0}^{H-1} \int_{\tau_k+h\delta}^{\tau_k+(h+1)\delta} \frac{4}{c}\mathcal{L}_{\boldsymbol{Q}_t}(\mu_t, \tilde{\boldsymbol{Q}}_{\tau_k+h\delta}\mu_t)\mathrm{d}t$$

$$\tag{39}$$

Since $\|\frac{s_\theta(\cdot;t+s)-s(\cdot;t)}{s(\cdot;t)}\|_\infty \leqslant \epsilon + Ls$. According to Lemma 3, and $\delta = t^*/H$,

$$\int_{\tau_k+h\delta}^{\tau_k+(h+1)\delta} \frac{4}{c}\mathcal{L}_{\boldsymbol{Q}_t}(\mu_t, \tilde{\boldsymbol{Q}}_{\tau_k+h\delta}\mu_t)\mathrm{d}t \leqslant \int_0^\delta \frac{4}{c}M(\epsilon + Ls)\mathrm{d}s = \frac{4M}{c}\left( \frac{\epsilon t^*}{H} + \frac{Lt^{*2}}{2H^2} \right) \tag{40}$$

Thus,

$$\int_{k(t^*+1)+1}^{(k+1)(t^*+1)} \mathsf{FI}_{\boldsymbol{Q}_t}(\mu_t\|\pi_t) \leqslant 2\Big(\mathsf{KL}(\tilde{p}_{k(t^*+1)+1}\|p_{k(t^*+1)+1}) - \mathsf{KL}(\tilde{p}_{(k+1)(t^*+1)}\|p_{(k+1)(t^*+1)})\Big) + \frac{4M}{c}(\epsilon t^* + \frac{Lt^{*2}}{2H})$$

Finally, taking summation over $k = 0, \ldots, K - 1$, and applying Lemma 1 for each likelihood step,

$$\sum_{k=0}^{K-1} \int_{k(t*+1)+1}^{(k+1)(t*+1)} \mathsf{FI}_{\boldsymbol{Q}_t}(\mu_t \| \pi_t) \leqslant 2\mathsf{KL}(\mu_0 \| \pi_0) + \frac{4M}{c}(\epsilon K t* + \frac{LKt*^2}{2H}).$$

Dividing by $Kt*$ on both sides,

$$\frac{1}{K} \sum_{k=0}^{K-1} \frac{1}{t*} \int_{k(t*+1)+1}^{(k+1)(t*+1)} \mathsf{FI}_{\boldsymbol{Q}_t}(\mu_t \| \pi_t) \leqslant \frac{2\mathsf{KL}(\mu_0 \| \pi_0)}{Kt*} + \frac{4M\epsilon}{c} + \frac{2MLt*}{cH}. \tag{41}$$

$\square$

## A.3  Potential function of split Gibbs samplers

As defined in Eq. (7), split Gibbs samplers draws samples from the augmented distribution

$$\pi(\mathbf{x}, \mathbf{z}; \eta) \propto \exp(-f(\mathbf{z}; \mathbf{y}) - g(\mathbf{x}) - D(\mathbf{x}, \mathbf{z}; \eta)).$$

The key requirement of split Gibbs samplers is that the potential function $D(\mathbf{x}, \mathbf{z}; \eta)$ satisfies

$$\lim_{\eta \to 0^+} D(\mathbf{x}, \mathbf{z}; \eta) = \infty, \forall \mathbf{x} \neq \mathbf{z}. \tag{42}$$

Or more precisely,

$$\lim_{\eta \to 0^+} \frac{\exp(-D(\mathbf{x}, \mathbf{z}; \eta))}{\int \exp(-D(\mathbf{x}, \mathbf{z}; \eta)) \mathrm{d}\mathbf{z}} = \delta_{\mathbf{x}}(\mathbf{z}). \tag{43}$$

Therefore,

$$\lim_{\eta \to 0^+} \pi^X(\mathbf{x}; \eta) \propto \lim_{\eta \to 0^+} \int p(\mathbf{x}) p(\mathbf{y} | \mathbf{z}) \exp(-D(\mathbf{x}, \mathbf{z}; \eta)) \mathrm{d}\mathbf{z}$$

$$= \int p(\mathbf{x}) p(\mathbf{y} | \mathbf{z}) \delta(\mathbf{x} - \mathbf{z}) \mathrm{d}z = p(\mathbf{x}) p(\mathbf{y} | \mathbf{x}). \tag{44}$$

Also, the similar derivation holds for $\pi^Z(\mathbf{z}; \eta)$. This result has also been shown in [46]. Combining this with Theorem 1, SGDD is guaranteed to sample from the true posterior distribution when $\eta$ goes to zero.

# B Ablation studies

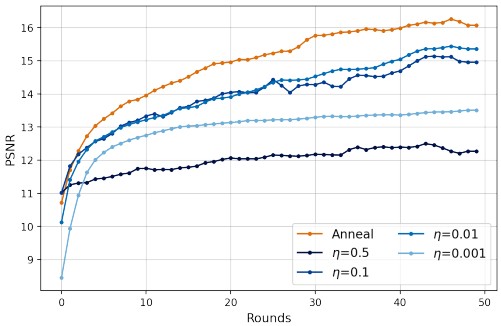

Figure 4: **Convergence speed of different noise schedules.** We compare our noise annealing scheduler $\{\eta_k\}_{k=1}^{K}$ (orange) to fixed noise schedules of various noise levels on the MNIST AND task. $y$-axis is the PSNR of variables $\mathbf{x}^{(k)}$ with respect to the ground truth. Our noise schedule converges faster than schedulers with fixed noises.

Figure 5: **Sampling quality vs. computing budget.** The $x$-axis indicates the number of function evaluations of the discrete diffusion, while the $y$-axis shows the PSNR metric. Experiments are done with 10 discretized MNIST samples on XOR and AND tasks.

## B.1 Effectiveness of the annealing noise schedule.

We investigate how the choice of noise schedule $\{\eta_k\}_{k=0}^{K}$ affects the performance of SGDD. In our main experiments, we use a geometric noise schedule from [29], defined as $\eta_k = \eta_{\min}^{k/K} \eta_{\max}^{1-k/K}$. We argue that an annealing noise schedule $\{\eta_k\}_{k=0}^{K}$ that gradually decays to $\eta_{\min} \approx 0$ allows SGDD to converge faster and more accurately to the posterior distribution.

To validate this hypothesis, we conduct an ablation study on the MNIST AND task. We run our method for $K = 50$ iterations and compare the annealing noise schedule with fixed noise schedules at different noise levels, ranging from $10^{-3}$ to $0.5$. We then compute the PSNR between $\mathbf{x}^{(k)}$ and the true underlying signal.

As shown in Fig. 4, running SGDD with the annealing noise schedule achieves the highest PSNR and also the fastest convergence speed. Moreover, when $\eta$ is large, SGDD converges quickly to a poor solution, whereas when $\eta$ is small, it converges slowly to an accurate solution. This finding aligns with our intuition that SGDD is most accurate with $\eta \to 0$, while sampling with $\eta > 0$ regularizes and simplifies the sampling problem.

This trade-off between convergence speed and accuracy results in a non-monotonic relationship between PSNR and fixed noise levels: PSNR first increases and then decreases with $\eta$. Therefore, using an annealing noise schedule helps provide both fast initial convergence and accurate posterior sampling as $\eta_k$ approaches 0.

**Theoretical insights.** Since Eq. (44) holds when $\eta \to 0$, a natural question is why we cannot fix a small $\eta$ to begin with.

The answer to this question is contained in Theorem 1. When fixing a small $\eta$, we also fix a small $t^*$ as $\sigma(t^*) = \eta$. Then the transition matrix $\boldsymbol{Q}_\tau$ ($\tau \in [0, t^*]$) will be ill-posed, in the sense that $\boldsymbol{Q}_\tau^{[i,j]}$ is either very large or very close to 0. In this case, the relative Fisher information $\mathsf{FI}_{\boldsymbol{Q}_\tau}(\mu_\tau \| \pi_\tau)$ becomes a poor criterion of measuring the divergence between $\mu_\tau$ and $\pi_\tau$. To see this, we note that the spectral gap $\gamma(\boldsymbol{Q}_\tau)$ has

$$\gamma(\boldsymbol{Q}_\tau) = \min_{\lambda \neq 0} -\mathrm{Re}\lambda(\boldsymbol{Q}_\tau) \geqslant N\epsilon, \tag{45}$$

where $N = |\mathcal{X}|$, if $\boldsymbol{Q}_\tau^{i,j} > \epsilon \geqslant 0$ for all $i \neq j$. By the logarithmic Sobolev inequality [3], we have that

$$\mathsf{FI}_{\boldsymbol{Q}_\tau}(\mu_\tau \| \pi_\tau) \geqslant \frac{n \min_{i \neq j} \boldsymbol{Q}_\tau^{[i,j]}}{2} \cdot \mathsf{KL}(\mu_\tau \| \pi_\tau) \tag{46}$$

Ignoring the errors in the score function and discretization, the KL divergence converges linearly, i.e.,

$$\mathsf{KL}(\mu_t \| \pi_t) \leqslant \exp\left(-\frac{n}{4} \int_0^t \min_{i \neq j} \boldsymbol{Q}_\tau^{[i,j]} \mathrm{d}\tau\right) \mathsf{KL}(\mu_0 \| \pi_0). \tag{47}$$

When $\eta_k$ is large, $\int_0^{t^*} \min_{i \neq j} \boldsymbol{Q}_\tau^{[i,j]} \mathrm{d}\tau$ is considerably large, because the probability ratio $p_t(\mathbf{x})$ is being smoothed by the uniform transition kernel, which makes $\boldsymbol{Q}_t^{[i,j]}$ large. However, as discussed above, $\int_0^{t^*} \boldsymbol{Q}_\tau^{[i,j]} \mathrm{d}\tau$ can be very close to 0 when $\eta_k$ is small, which makes the convergence slow.

## B.2 Time efficiency.

We run SGDD with different configurations to evaluate its performance under different computing budgets. We control the number of iterations, $K$, and the number of unconditional sampling steps $T$ in each prior sampling step, so the number of function evaluations (NFE) of the discrete diffusion model is $KT$. We evaluate SGDD with NFEs ranging from 100 to 2k. The detailed configuration is specified in Table 6.

We present a plot in Fig. 5 showing the tradeoff between sample quality and computing budgets in terms of NFEs. SGDD is able to generate high-quality samples with a reasonable budget comparable to unconditional generation. Note that we calculate the NFEs for SVDD and SMC with the number of Monte Carlo samples $\mathrm{mc} = 1$ and the number of Euler steps $H = 1$, while some results in our experiments are done with larger $\mathrm{mc}$ and $H$ for better performance. We report both the number of total NFEs and the number of sequential NFEs that decide the cost of the algorithm when it is fully parallelized.

Table 6: **Configurations of SGDD and baseline methods.** For DPS and SVDD-PM, we calculate both the total NFE and the sequential NFE. The numbers are collected from the MNIST XOR task as an example. The runtime is amortized on a batch size of 10 samples with one NVIDIA A100 GPU.

| Configuration | # Iterations | Prior sampling steps | NFE (total/sequential) | Runtime (sec/sample) |
|---|---|---|---|---|
| DPS | 100 | - | 102400/100 | 820 |
| SVDD-PM ($M = 20$) | 128 | - | 2560/128 | 16 |
| SMC ($M = 20$) | 128 | - | 5120/256 | 21 |
| SGDD-50 | 25 | 2 | 50 | 4 |
| SGDD-100 | 25 | 4 | 100 | 6 |
| SGDD-200 | 40 | 5 | 200 | 8 |
| SGDD-400 | 40 | 10 | 400 | 10 |
| SGDD-1k | 50 | 20 | 1000 | 13 |
| SGDD-2k | 100 | 20 | 2000 | 20 |

## B.3 Comparison to sampling without prior.

We evaluate the value of discrete diffusion priors in solving discrete inverse problems by comparing our method to the Metropolis-Hastings algorithm without a diffusion prior. We pick the AND problem on discretized MNIST as an example, in which we choose $\gamma D$ random pairs of pixels and compute their AND values as measurement $\mathbf{y}$. Fig. 6 shows the reconstructed samples of SGDD and direct MCMC sampling under different levels of measurement sparsity. As demonstrated in the figure, the discrete diffusion model complements the missing information when the measurement is sparse (e.g., $\gamma = 2, 4, 8$), underscoring the importance of discrete diffusion priors in SGDD for solving discrete inverse problems.

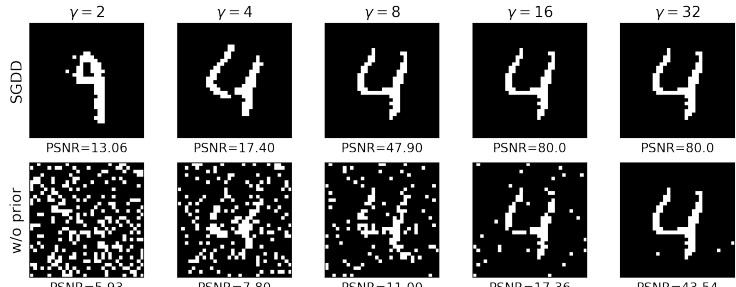

Figure 6: **Comparison of SGDD and direct sampling without data prior.** The PSNR values are computed with 10 test samples. SGDD replenishes the missing information when the measurement is sparse.

## C   Experimental details

### C.1   Pretrained Models

We learn prior distributions for each dataset using SEDD [29] discrete diffusion models. We use the SEDD small architecture with around 90M parameters for all experiments, and the models are trained with AdamW [28] with batch size 32 and a learning rate of $3 \times 10^{-4}$.

### C.2   Baseline Methods

#### C.2.1   DPS

DPS [7] is designed to solve general inverse problems with a pretrained (continuous) diffusion model. It performs posterior sampling from $p(\mathbf{x}|\mathbf{y})$ by modifying the reverse SDE

$$\mathrm{d}\mathbf{x}_t = -2\dot{\sigma}_t\sigma_t\nabla_{\mathbf{x}_t}\log p(\mathbf{x}_t|\mathbf{y})\mathrm{d}t + \sqrt{2\dot{\sigma}_t\sigma_t}\mathrm{d}t \tag{48}$$

$$= -2\dot{\sigma}_t\sigma_t(\nabla_{\mathbf{x}_t}\log p(\mathbf{x}_t; \sigma_t)) + \nabla_{\mathbf{x}_t}\log p(\mathbf{y}|\mathbf{x}_t))\mathrm{d}t + \sqrt{2\dot{\sigma}_t\sigma_t}\mathrm{d}t. \tag{49}$$

To estimate the intractable guidance term $\nabla_{\mathbf{x}_t}\log p(\mathbf{y}|\mathbf{x}_t)\mathrm{d}t$, DPS proposes to approximate it with $p(\mathbf{y}|\mathbf{x}_t) \approx p(\mathbf{y}|\mathbb{E}[\mathbf{x}_0|\mathbf{x}_t])$. The guidance term is thus approximated by

$$\nabla_{\mathbf{x}_t}\log p(\mathbf{y}|\mathbf{x}_t) \approx -\nabla_{\mathbf{x}_t}\frac{\|\boldsymbol{G}(\boldsymbol{D}_\theta(\mathbf{x}_t, \sigma_t)) - \mathbf{y}\|_2^2}{2\sigma_{\mathbf{y}}^2}, \tag{50}$$

where $\boldsymbol{D}_\theta$ is a one-step denoiser using the pretrained diffusion model, and the measurement is assumed to be $\mathbf{y} = \boldsymbol{G}(\mathbf{x}) + \mathbf{n}$ with $\mathbf{n} \sim \mathcal{N}(0, \sigma_{\mathbf{y}}^2\boldsymbol{I})$.

However, DPS is not directly applicable to inverse problems in discrete-state spaces since propagating gradients through $\boldsymbol{G}$ and $\boldsymbol{D}_\theta$ in Eq. (50) is impossible. Therefore, we consider the counterpart of DPS in discrete spaces. We modify the continuous-time Markov chain of the discrete diffusion model by

$$\frac{\mathrm{d}p_{T-t}}{\mathrm{d}t} = \boldsymbol{Q}_t^{\mathbf{y}}p_{T-t}, \tag{51}$$

in which

$$\boldsymbol{Q}_t^{\mathbf{y}\,[i,j]} = \frac{p_{T-t}(\mathbf{x}_i|\mathbf{y})}{p_{T-t}(\mathbf{x}_j|\mathbf{y})} = \frac{p_{T-t}(\mathbf{x}_i)}{p_{T-t}(\mathbf{x}_j)}\frac{p_{T-t}(\mathbf{y}|\mathbf{x}_i)}{p_{T-t}(\mathbf{y}|\mathbf{x}_j)} = \boldsymbol{Q}_t^{[i,j]} \cdot \frac{p_{T-t}(\mathbf{y}|\mathbf{x}_i)}{p_{T-t}(\mathbf{y}|\mathbf{x}_j)} \tag{52}$$

Similar ideas are applied to classifier guidance for discrete diffusion models [32], where the matrix $\boldsymbol{R}_t^{\mathbf{y}} = [\frac{p_{T-t}(\mathbf{y}|\mathbf{x}_i)}{p_{T-t}(\mathbf{y}|\mathbf{x}_j)}]_{i,j}$ is called a guidance rate matrix. We compute $\boldsymbol{R}_t^{\mathbf{y}}$ at $\mathbf{x}_j$-column by enumerating every neighboring $\mathbf{x}_i$ and calculating $\frac{p_t(\mathbf{y}|\mathbf{x}_i)}{p_t(\mathbf{y}|\mathbf{x}_j)}$ for each $\mathbf{x}_i$. The discrete version of DPS can be summarized by

$$\frac{\mathrm{d}p_{T-t}}{\mathrm{d}t} = \boldsymbol{Q}_t\boldsymbol{R}_t^{\mathbf{y}}p_{T-t}. \tag{53}$$

However, the discrete version of DPS is very time-consuming, especially when the vocabulary size is large, for it enumerates $(N-1) \times D$ number of neighboring $\mathbf{x}$ when computing $\boldsymbol{R}_t^{\mathbf{y}}$. We find it slow for binary MNIST ($N = 2$) and DNA generation tasks ($N = 4$) but unaffordable for monophonic music generation ($N = 129$).

### C.2.2 SVDD

SVDD [27] aims to sample from the distribution $p^\beta(\mathbf{x}_0) \propto p(\mathbf{x}_0) \exp(\beta r(\mathbf{x}_0))$, which is equivalent to the regularized MDP problem:

$$p^\beta(\mathbf{x}_0) = \arg\max_\pi \mathbb{E}_{\mathbf{x}_0 \sim \pi} r(\mathbf{x}_0) - \mathsf{KL}(\pi \| p)/\beta \tag{54}$$

They calculate the soft value function as

$$v_t(\mathbf{x}_t) = \log \mathbb{E}_{\mathbf{x}_0 \sim p(\mathbf{x}_0|\mathbf{x}_t)}[\exp(\beta r(\mathbf{x}_0))]/\beta \tag{55}$$

and propose to sample from the optimal policy

$$p_t^\star(\mathbf{x}_t|\mathbf{x}_{t+1}) \propto p_t(\mathbf{x}_t|\mathbf{x}_{t+1}) \exp(\beta v_t(\mathbf{x}_t)). \tag{56}$$

In time step $t$, SVDD samples a batch of $M$ particles from the unconditional distribution $p_t(\mathbf{x}_t|\mathbf{x}_{t+1})$, and conduct importance sampling according to $\exp(\beta v_t(\mathbf{x}_t))$.

Although [27] is initially designed for guided diffusion generation; it also applies to solving inverse problems by carefully choosing reward functions. We consider $r(\mathbf{x}) = -\|\boldsymbol{G}(\mathbf{x}) - \mathbf{y}\|_0/\sigma_{\mathbf{y}}$ and $\beta = 1$, so that it samples from the posterior distribution $p(\mathbf{x}|\mathbf{y})$. As recommended in [27], we choose $\beta = \infty$ ($\alpha = 0$ in their notation) in practice, so the importance sampling reduces to finding the particle with the maximal value in each iteration. We use SVDD-PM, a training-free method provided by [27] in our experiments. It approximates the value function by $v_t(\mathbf{x}_t) = r(\hat{\mathbf{x}}_0(\mathbf{x}_t))$, where $\hat{\mathbf{x}}_0(\mathbf{x}_t)$ is an approximation of $\mathbb{E}[\mathbf{x}_0|\mathbf{x}_t]$. In practice, we find that approximating $\hat{\mathbf{x}}_0(\mathbf{x}_t)$ by Monte Carlo sampling with a few-step Euler sampler achieves slightly better results. We use 3 Monte Carlo samples to estimate $v_t(\mathbf{x}_t)$ in our experiments.

### C.2.3 SMC

Sequential Monte Carlo (SMC) methods evolve multiple particles to approximate a series of distributions, eventually converging to the target distribution. Specifically, in our experiments, we implement the SMC method to sample from $p(\mathbf{x}_t|\mathbf{y})$ for $t = T, T - 1 \dots, 0$, using the unconditional discrete diffusion sampler $p(\mathbf{x}_t|\mathbf{x}_{t+1})$ as the proposal function.

We maintain a batch of $M = 20$ particles $\{\mathbf{x}^{[m]}\}$. At time $t$, we sample $\mathbf{x}_t^{[m]} \sim p(\mathbf{x}_t|\mathbf{x}_{t+1}^{[m]})$ by the pretrained discrete diffusion model and estimate the likelihood $p(\mathbf{y}|\mathbf{x}_t^{[m]}) = \mathbb{E}_{\mathbf{x}_0 \sim p(\mathbf{x}_0|\mathbf{x}_t^{[m]})} p(\mathbf{y}|\mathbf{x}_0)$ by Monte Carlo sampling. We then resample the particles $\{\mathbf{x}_t^{[m]}\}$ according to their weights $w_t^{[m]} = p(\mathbf{y}|\mathbf{x}_t^{[m]})/p(\mathbf{y}|\mathbf{x}_{t+1}^{[m]})$. In practice, we find that we have to carefully tune a hyperparameter $\beta$, where $w_t^\beta$, or otherwise the resampling step can easily degenerate to $\arg\max$ or uniform random sampling.

### C.3 Hyperparameters

We use an annealing noise schedule of $\eta_k = \eta_{\min}^{k/K} \eta_{\max}^{1-k/K}$ with $\eta_{\min} = 10^{-4}$ and $\eta_{\max} = 20$. We run SGDD for $K$ iterations. In each likelihood sampling step, we run Metropolis-Hastings for $T$ steps, while in each prior sampling step, we run a few-step Euler discrete diffusion sampler with $H$ steps. The hyperparameters used for each experiment are listed in Table 7. We also include the dimension of data spaces $\mathcal{X}^D$ for each experiment in Table 7, where $|\mathcal{X}| = N$.

Table 7: **Hyperparameters used in each experiment.**

|  | Synthetic | DNA design | MNIST XOR | MNIST AND | Music infilling |
|---|---|---|---|---|---|
| Metropolis-Hastings $T$ | 10 | 200 | 2000 | 5000 | 5000 |
| SGDD iterations $K$ | 10 | 50 | 50 | 100 | 100 |
| Euler sampler $H$ | 20 | 20 | 20 | 20 | 20 |
| Sequence length $D$ | 2/5/10 | 200 | 1024 | 1024 | 256 |
| Vocab size $N$ | 50 | 4 | 2 | 2 | 129 |

# D  Additional Experimental Results

## D.1  Solving image restoration problems in discrete latent space.

To demonstrate the performance of SGDD on higher-dimensional datasets, we conduct an additional experiment on FFHQ 256 with a vector-quantized diffusion model. Specifically, we use a pretrained VQVAE model from [13] and train a discrete diffusion model in the discrete latent space. We test our algorithm on the super-resolution task ($\times 4$) using 100 images from the test dataset. While existing methods fail to solve this high-dimensional task with a complicated forward model that includes a latent space decoder (the cost of DPS is unaffordable), our method is able to reconstruct the underlying images with high PSNR values.

While the latent codes of VQVAE lie in a discrete-state space, they are embedded in an underlying Euclidean space, which makes this problem different from solving inverse problems in categorical distributions without ordinal information. A potential way to speed up and boost the performance of SGDD in solving image restoration problems with VQ-diffusion is to replace the Metropolis-Hastings in discrete latent codes with Langevin dynamics in the continuous embedding space of latent codes. We leave the exploration of this idea as a future direction.

Table 8: Quantitative results for the super-resolution ($\times 4$) task on FFHQ 256 dataset.

|  | PSNR ↑ | LPIPS ↓ |
|---|---|---|
| SVDD-PM | $12.08_{\pm 0.39}$ | $0.594_{\pm 0.061}$ |
| SGDD | $\mathbf{25.85}_{\pm 0.39}$ | $\mathbf{0.288}_{\pm 0.039}$ |

Table 9: Class-conditioned generation on MNIST dataset.

|  | MSID ↓ | Accuracy % |
|---|---|---|
| SMC | 39.13 | 45.0 |
| SGDD | **14.59** | **98.5** |

## D.2  More results for inverse problems on discretized image.

**Qualitative results.** Here we present the visualization of the reconstructed samples of MNIST from their XOR measurements.

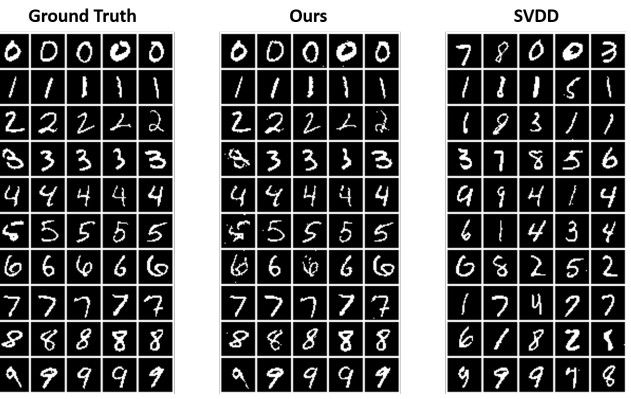

Figure 7: **Sampling results of the XOR task on the discretized MNIST dataset.** SGDD faithfully recovers the structural information of the ground truth signal.

**Approximate distribution divergence on class-conditioned generation.** Since a classifier on MNIST data can also be viewed as a forward model, SGDD can also be directly applied to sample digit-conditioned samples. Moreover, while the posterior distribution for a general inverse problem is inaccessible, we can approximate the posterior distribution of class-conditioned generation by a digit-specific prior distribution.

Specifically, we define the posterior distribution as proportional to $p(\mathbf{x})p(-\beta\ell(\boldsymbol{G}(\mathbf{x}), \mathbf{y}))$, where $\ell$ is the cross-entropy loss between the predicted class probability and target class label $\mathbf{y}$. We let $\mathbf{y} = 0$ to guide diffusion models to generate digits $0$, and extract all digits $0$ in the dataset as the approximated true posterior distribution. We train another classifier with a different random seed to evaluate the generated samples. To measure the divergence of the generated samples from the posterior distribution, we extract the features of MNIST before the last layer of a CNN classification network, and compute the multi-scale intrinsic distance (MSID) score.

Experimental results in Table 9 show that SGDD effectively samples from the true posterior distribution, as it achieves a low MSID score and a high class accuracy of $98.5\%$.

### D.3 More experimental results on the synthetic dataset

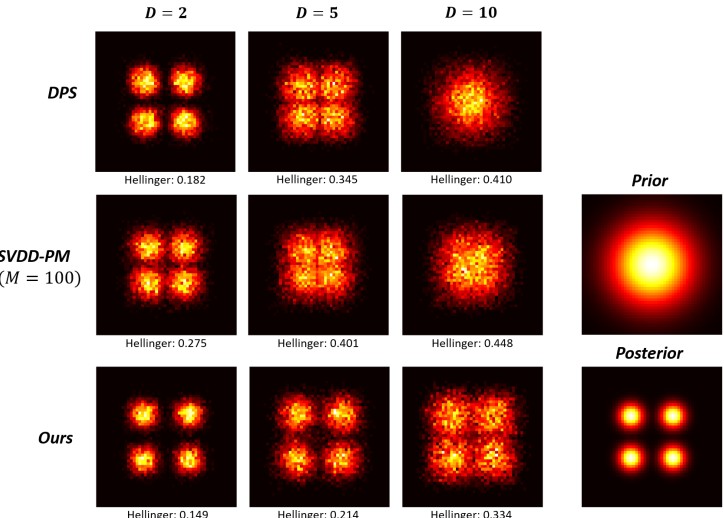

Figure 8: **Empirical distributions of various sampling algorithms on the synthetic dataset.** Heatmaps are drawn with 10k samples for sequence lengths of $D = 2, 5$ and $10$. We project the sequences to the first two dimensions for ease of visualization.

# E Discussions

## E.1 Limitations and future directions

Our method is built on discrete diffusion models with a uniform transition matrix. It is unclear how our method can be applied to other types of discrete diffusion models, such as masked diffusion models. Moreover, our method calls the forward model (for inverse problems) or the reward function multiple times while running Metropolis-Hastings in likelihood steps, which potentially increases the computational overhead when the forward model or reward function is complicated. We leave these issues as possible future extensions.

## E.2 Boarder Impacts

We anticipate that SGDD offers a clean and principled framework for posterior sampling using discrete diffusion models. SGDD tackles these problems by leveraging a discrete diffusion model as a general denoiser in the finite space, which models the prior distribution with rich data. However, we note that it could be the case that SGDD will generate biased samples, if the discrete diffusion model is itself trained on biased data. Therefore, caution should be exercised when using SGDD in certain sensitive scenarios.

