# OpenReview forum: "Split Gibbs Discrete Diffusion Posterior Sampling"
_NeurIPS.cc/2025/Conference — NeurIPS 2025 poster_

### Official Review · Reviewer_1thq · 2025-06-27

**Clarity:** 4
**Significance:** 3
**Originality:** 3
**Rating:** 5
**Confidence:** 3

**Summary:**

- Method for sampling from posterior distribution $p(x|y)$ and reward tilted distribution.
    - Introduce an auxiliary variable $z$ and an augmented distribution $\pi$
    - Both marginal of $\pi$ converge to the posterior distribution.
    - Sampling is done by alternating a likelihood sampling step and a prior sampling step.
- They adapt prior work  to discrete diffusion.
- Prove the convergence of their method
- Substantial experiments across challenging tasks

**Questions:**

See weakness

**Ethical Concerns:**

["NO or VERY MINOR ethics concerns only"]

**Final Justification:**

I believe this work offers an interesting method with solid experiments to sample from discrete models. It is experimentally and theoretically strong and deserves in my opinion to be accepted.

**Limitations:**

Yes

**Paper Formatting Concerns:**

No formatting concerns

**Quality:**

3

**Strengths And Weaknesses:**

### Strengths

- The paper is well written and easy to follow.
Figure 1 illustrates the method well and helps understanding the paper.
- The method does not require gradients w.r.t the likelihood.
- The experiments are substantial, challenging and showcases the performance of the method.
- The theory is convincing and presented clearly.
- It appears the method does not need any additional training.
- The insights regarding noise schedule selection for sampling are valuable and well-explained.

### Weaknesses

- The framework appears designed specifically for uniform transition kernels. Could the authors clarify this limitation? If confirmed, what prevents extension to more general kernels? While absorbing kernels have inherent limitations, they currently represent the state-of-the-art, and demonstrating compatibility would significantly strengthen the work. Additionally, recent literature explores more promising kernels beyond masked processes, and generalization to these newer approaches would enhance the method's broader applicability.

#### Theoretical concerns

I have not read the proof in details but from what I have seen I do not see any theoretical concerns.

---

> ### Author Rebuttal · Authors · 2025-07-31
>
> > 1. Is SGDD limited to discrete diffusion models with uniform transition kernels? What prevents extension to more general kernels?
>
> Yes, SGDD only considers discrete diffusion models with uniform transition kernel. Recall that we design a potential function $D$ in Equation (10) to make Equation (9) equivalent to a partial prior sampling from discrete diffusion models. The primary challenge of extending SGDD to other types of discrete diffusion model is to find another potential function that relates Equation (9) to prior sampling. However, masked diffusion models requires operating on an additional [MASK] token, on which the likelihood function is not defined, making this extension a nontrivial task. Nonetheless, we believe it is still possible if we carefully specify the likelihood function on sequences with [MASK] tokens as well, but we will leave this as an interesting future exploration.

---

### Official Review · Reviewer_z3tY · 2025-06-30

**Clarity:** 2
**Significance:** 3
**Originality:** 3
**Rating:** 4
**Confidence:** 4

**Summary:**

The paper proposes an adaptation of the PnPDM split Gibbs sampler ([1]) to sampling from conditional distributions that use a discrete diffusion generative models as priors. To do so, it proposes using a different coupling potential for the two variables of the split Gibbs that makes the prior step of the Gibbs sampler be the equivalent of the backward part of the generative model, much like the standard gaussian coupling potential does for the continuous split Gibbs sampler in [1]. It then states a theoretical results for fixed $eta$ law of the Gibbs sampler, that is heavily inspired by the theoretical result of [1]. It then proceeds to a numerical verification of the proposed algorithm in several benchmarks.

**Questions:**

See weaknesses for the questions that I had, manly on the numerics part. If some of the writing is improved and details to the numerics are added I am willing to increase my grade.

**Ethical Concerns:**

["NO or VERY MINOR ethics concerns only"]

**Final Justification:**

The authors properly addressed my concerns and promised to add more details of the experimental section to the revised version. Therefore, I am inclined to suggest accepting the paper.

**Limitations:**

yes

**Quality:**

2

**Strengths And Weaknesses:**

Strengths:
The paper proposes an interesting way of adapting PnPDM to the discrete diffusion case. The approach is sound and the adaptation of the coupling potential is appropriate. The results about the split Gibbs are described in a detailed and precise fashion.

Weaknesses:
1. The claims in the paper concerning SMC-based methods are highly biased. Namely, lines 190-192. SMC guarantees hold for finite number of particles and I do not see why the guarantees from theorem 1 are inherently better than SMC. Furthermore, the role of the potential functions on SMC can be equated to the choices of $eta$ in the split Gibbs mechanism. Furthermore, most SMC guarantees regard the target posterior distribution and not a $eta$-version as in theorem 1. Therefore, this claim, while not at all necessary to the paper, which could justify their sampler by other merits of the sampler in itself, lead to a biased presentation of the current literature in the topic, which is not acceptable, specially in a prestigious venue as NeurIPS.

2. Numerics pt1: Do the authors try to equate the NFE for all the samplers in table 2 and all section 4? This is on of the only ways of establishing "fair" comparison between different samplers, so one would expect so. But, as far as I'm concerned, even in the appendix I cannot find the number of particles used in the SMC approach (except for table 6, which is a different type of experiment).

3. Numerics pt2: Several other small details are missing in the numerics, namely what is exactly the function $G$ in the Monophonic Music problem.

---

> ### Author Rebuttal · Authors · 2025-07-31
>
> > 1. Clarifications on the claims about SMC-based methods.
>
> Thanks for pointing this out. The claims made in lines 190-192 were only targeting the baseline methods SVDD [1] and TDS [2]. SVDD estimates a value function of noisy samples during diffusion sampling, and the guarantees of TDS hold only when the number of particles goes to infinity. However, the reviewer is right that these claims are not true for SMC-based methods in general. We apologize for causing these confusions, and we will correct this claim in our revised version.
>
> Moreover, we further include a more recent SMC-based method from Feynman-Kac (FK) steering [3], with MAX potential function (i.e. the best value along the sampling history). We use $M=20$ particles for all SMC-based experiments. Following this experiment, we will also include a more comprehensive literiture review in Section 2.2 to discuss recent advancements of SMC-based methods. As we see in the results below, while FK-MAX performs slightly better than TDS, our SGDD method significantly outperforms both SMC-based methods.
>
> | DNA | Pred-Activity (median) | ATAC Acc | Log-likelihood |
> | --  | --| -- | -- |
> | TDS [2] | 4.64 | 45.3% | -257|
> | FK-MAX [3] | 4.90 | 47.1% | -243 |
> | SGDD | 9.14 | 93.0% | -261|
>
> | MNIST-XOR | PSNR | Accuracy |
> | -- | -- | -- |
> | SMC | 10.05 | 27.8% |
> | FK-MAX [3] | 11.20 | 34.7% |
> | SGDD | 20.17 | 91.2% |
>
> > 2. Equating number of functional evaluations.
>
> We use the same configurations (number of particles $M=20$ and NFEs) for SMC and SVDD in all experiments, as shown in Table 6 of Appendix B, resulting in 5120 NFEs for SMC methods. The configurations for SGDD is described in Table 7, with NFEs ($=H\cdot K$) ranging from 200 to 2000, consistently lower than the NFEs used by baseline methods. While the the functional evaluations of SMC-based methods can be parallelized, we compute and compare the runtime of each method (in Table 6) on MNIST-XOR as an example.
>
> > 3. Missing details on monophonic music task.
>
> Thanks for pointing this out! The forward model $G$ is an operator that randomly masks $\gamma = 40\\%$ or $60\\%$ of the notes. We will make this clear in our revision.
>
> [1] Li et al.  Derivative-free guidance in continuous and discrete diffusion models with soft value-based decoding, 2024.
>
> [2] Wu et al. Practical and asymptotically exact conditional sampling in diffusion models, 2024.
>
> [3] Singhal et al. A general framework for inference-time scaling and steering of diffusion models, 2025.

---

> > ### Comment · Reviewer_z3tY · 2025-08-05
> >
> > I thank the authors for addressing my concerns, namely by providing more details for the numerical part.

---

### Official Review · Reviewer_f55Z · 2025-07-03

**Clarity:** 3
**Significance:** 3
**Originality:** 3
**Rating:** 5
**Confidence:** 2

**Summary:**

This paper presents a new diffusion model sampling technique that can be used on discrete target distributions.
They augment the space by introducing an auxiliary random variable, $z$, that appears in a "regularization potential" term, $D(.)$, controlling the divergence of the intermediate distributions from the target. The augmented posterior is approximated by Gibbs sampling and via updating the original and auxiliary RVs, $x$ and $z$, iteratively. In this context this technique is called "Split Gibbs" sampling and is already applied to continuous settings. To apply Split Gibbs sampling to the discrete setting, the current work specifies a new potential function which leads to the new techque, namely Split Gibbs Discrete Diffusion sampling (SGDD). The convergence of the proposed algorithm is proved w.r.t. the Fisher divergecne.
SGDD is compared with multiple algorithms for discrete diffusion posterior sampling across multiple domains. The reported results show that on these models, SGDD outperforms the existing baselines.

**Questions:**

1. In the theorem, the convergence is w.r.t. the Fisher divergence, but in Table 2, the performance is measured w.r.t. Hellinger distance. Could you also report the Fisher divergence as well as the KL divergence from the target?

2. Could you provide a link to the code?

**Ethical Concerns:**

["NO or VERY MINOR ethics concerns only"]

**Limitations:**

This is a theoretical work with no direct social impact.

**Paper Formatting Concerns:**

No concerns

**Quality:**

3

**Strengths And Weaknesses:**

Strengths:
The paper is well-written; the preliminaries and related work are sufficiently discussed; the SGDD algorithm is explained clearly and compared against several relevant state-of-the-art algorithms on various models.

Weaknesses:
Minor comments on the notation:
1. In lines 68-70 and 140, I assume $\mathbf{11}^T$?  represents an $N \times N$ matrix of ones? This notation is a bit confusing and should be clarified.
2. Define $\dot{\sigma}_t$, (I suppose it denotes $d \sigma_t /dt$).

Please also see the Section "Questions" below.

---

> ### Author Rebuttal · Authors · 2025-07-31
>
> > 1. Ambiguities in notations.
>
> Yes, $\mathbf{1}\mathbf{1}^T$ means a matrix of all ones, and $\dot \sigma\_t$ is defined as $\mathrm d\sigma\_t/\mathrm dt$. We will clarify these notations in the revised version.
>
> > 2. Providing Fisher divergence and KL divergence in table 2.
>
> Thanks for the suggestion. The relative Fisher divergence is considered for the purpose of theoretical analysis. It is hard to evaluate relative Fisher divergence, because it is defined with respective to the rate matrix $\mathbf Q\_t$. However, we provide the KL divergence for the experiments on synthetic data:
>
> | KL divergence | D=2 | D=5 | D=10 |
> | -------- | -------- | -------- | --- |
> | SVDD-PM (M=20) |  0.374 | 0.826 | 1.087 |
> | SMC (TDS) | 0.158 | 0.567 | 0.909 |
> | DPS  | 0.117 | 0.586 | 0.836 |
> | SGDD | 0.072 | 0.204 | 0.560 |
>
> > 3. Could you provide a link to the code?
>
> Thanks for your interest! Please check our code in the supplementary file.

---

> > ### Comment · Reviewer_f55Z · 2025-08-07
> >
> > I thank the authors for the clarification and new experimental results. I keep my score.

---

### Official Review · Reviewer_am6s · 2025-07-08

**Clarity:** 2
**Significance:** 3
**Originality:** 2
**Rating:** 3
**Confidence:** 5

**Summary:**

This paper proposed Split Gibbs Discrete Diffusion (SGDD) for posterior sampling in discrete-state spaces, which generalizes the methodology proposed in [1,2,3] that applies continuous diffusion models to posterior sampling problems in continuous-state spaces to discrete distributions. Similar to the procedure developed in [1,2,3], the method consists of two alternative steps. One step is the likelihood sampling step, which is performed via standard Markov Chain Monte Carlo (MCMC) method. The other step is the prior sampling step, which is performed by running a discrete diffusion model. Both theoretical analysis of the proposed algorithm and empirical experiments on a wide range of tasks like DNA design, discrete image inverse problems and music infilling are included to justify the effectiveness of the proposed method.

**Questions:**

Regarding the theoretical part presented in the manuscript, would it be possible for the authors to comment on the relation between the relative Fisher information FI defined here with the score entropy loss proposed in [21]? It seems to the reviewer that the two forms are akin and highly related. Instead of adopting the stronger assumption on the error between the true score function and the learnt score function with respect to the $l_{\infty}$ norm, the reviewer thinks that it is probably better to assume an upper bound on the training loss directly, just as what has been done in [13]. Moreover, it seems that the most crucial part in the theoretical results derived here is essentially bounding the KL divergence between two continuous-time markov chains (CTMCs) $\frac{\partial}{\partial t}\mu_t = Q_t \mu_t$ and $\frac{\partial}{\partial t}\widehat{\mu}_t = \widehat{Q}_t \widehat{\mu}_t$ via the discrepancy between $Q_t$ and $\widehat{Q}_t$, which highly relates to the Girsanov Theorem for CTMCs (See for instance Theorem III.5.34 in the book [11] or Theorem 3.3 in [13] for the case of discrete diffusion models). Hence, the authors might consider citing these related work and briefly comment on the relation between all different derivations.

In addition, to the best of the reviewer's knowledge, recent work on solving inverse problems via continuous diffusion models have also adopted advanced solvers like [19] to speed up the inference process. Would it be possible for the authors to comment on whether it would be possible for SGDD to be accelerated via more efficient inference methods (for discrete diffusion models) like distillation [22] or high-order solvers [16]?

**Ethical Concerns:**

["NO or VERY MINOR ethics concerns only"]

**Final Justification:**

After reviewing the authors' rebuttal, the reviewer is satisfied with nearly all of their responses, which range from literature review, theoretical analysis and large-scale experiments. However, the reviewer still has a few concerns for the methodology and theoretical analysis presented in the paper.

For the methodology, the reviewer thinks that its novelty is a bit limited as it seems to be a simple generalization of existing work [1,2] (in the References above) from the continuous domain to the discrete domain. For the theoretical analysis, the reviewer finds it to be a bit limited, as it assume a fixed eta parameter,  which is not the case in practical applications.

Based on the two points as well as all question raised in the review above, the reviewer is recommending "Borderline Reject" by raising the score from 2 to 3.

**Limitations:**

The reviewer is satisfied with the limitations that the authors briefly discussed at the end of the manuscript. For instance, the authors mentioned that currently the methodology can only be used for discrete diffusion models with uniform transition matrix. Hence, one of the future directions is to investigate whether the methodology can be generalized to other forms of discrete diffusion models like the masked discrete diffusion model with absorbing transition matrix. However, this is indeed an important issue as masked discrete diffusion models (akin to autoregressive models) are more widely adopted nowadays in other tasks.

Overall, the reviewer acknowledges that the authors address an important problem and offer meaningful contributions. However, the manuscript’s quality (presentation, literature review, experiments and theoretical analysis) probably requires further improvement before being considered for publication at a top-tier machine learning venue.

References:

[1] Wu, Z., Sun, Y., Chen, Y., Zhang, B., Yue, Y., & Bouman, K. (2024). Principled probabilistic imaging using diffusion models as plug-and-play priors. Advances in Neural Information Processing Systems (NeurIPS), 37, 118389-118427.

[2] Xu, X., & Chi, Y. (2024). Provably robust score-based diffusion posterior sampling for plug-and-play image reconstruction. Advances in Neural Information Processing Systems (NeurIPS), 37.

[3] Coeurdoux, F., Dobigeon, N., & Chainais, P. (2024). Plug-and-play split Gibbs sampler: embedding deep generative priors in Bayesian inference. IEEE Transactions on Image Processing.

[4] Singhal, R., Horvitz, Z., Teehan, R., Ren, M., Yu, Z., McKeown, K., & Ranganath, R. (2025). A general framework for inference-time scaling and steering of diffusion models. arXiv preprint arXiv:2501.06848.

[5] Chen, H., Ren, Y., Min, M. R., Ying, L., & Izzo, Z. (2025). Solving inverse problems via diffusion-based priors: An approximation-free ensemble sampling approach. arXiv preprint arXiv:2506.03979.

[6] Skreta, M., Akhound-Sadegh, T., Ohanesian, V., Bondesan, R., Aspuru-Guzik, A., Doucet, A., ... & Neklyudov, K. (2025). Feynman-kac correctors in diffusion: Annealing, guidance, and product of experts. arXiv preprint arXiv:2503.02819.

[7] Trippe, B. L., Yim, J., Tischer, D., Baker, D., Broderick, T., Barzilay, R., & Jaakkola, T. (2022). Diffusion probabilistic modeling of protein backbones in 3d for the motif-scaffolding problem. In The Eleventh International Conference on Learning Representations (ICLR), 2023.

[8] Kim, S., M. Kim, and D. Park (2025). Test-time alignment of diffusion models without reward over-optimization. In The Thirteenth International Conference on Learning Representations (ICLR), 2025.

[9] Yoon, T., Min, Y., Yeo, K., & Sung, M. (2025). $\Psi $-Sampler: Initial Particle Sampling for SMC-Based Inference-Time Reward Alignment in Score Models. arXiv preprint arXiv:2506.01320.

[10] Uehara, M., Zhao, Y., Wang, C., Li, X., Regev, A., Levine, S., & Biancalani, T. (2025). Inference-Time Alignment in Diffusion Models with Reward-Guided Generation: Tutorial and Review. arXiv preprint arXiv:2501.09685.

[11] Jacod, J., & Shiryaev, A. (2013). Limit theorems for stochastic processes (Vol. 288). Springer Science & Business Media.

[12] Chen, H., & Ying, L. (2024). Convergence analysis of discrete diffusion model: Exact implementation through uniformization. arXiv preprint arXiv:2402.08095.

[13] Ren, Y., Chen, H., Rotskoff, G. M., and Ying, L. How discrete and continuous diffusion meet: Comprehensive analysis of discrete diffusion models via a stochastic integral framework. In The Thirteenth International Conference on Learning Representations (ICLR), 2025

[14] Zhang, Z., Chen, Z., & Gu, Q. (2024). Convergence of score-based discrete diffusion models: A discrete-time analysis. In The Thirteenth International Conference on Learning Representations (ICLR), 2025

[15] Liang, Y., Huang, R., Lai, L., Shroff, N., & Liang, Y. (2025). Absorb and Converge: Provable Convergence Guarantee for Absorbing Discrete Diffusion Models. arXiv preprint arXiv:2506.02318.

[16] Ren, Y., Chen, H., Zhu, Y., Guo, W., Chen, Y., Rotskoff, G. M., Tao, M., and Ying, L. Fast solvers for discrete diffusion models: Theory and applications of high-order algorithms. arXiv preprint arXiv:2502.00234, 2025a

[17] Murata, N., Lai, C. H., Takida, Y., Uesaka, T., Nguyen, B., Ermon, S., & Mitsufuji, Y. (2024). G2D2: Gradient-guided Discrete Diffusion for image inverse problem solving. arXiv preprint arXiv:2410.14710.

[18] Karras, T., Laine, S., & Aila, T. (2019). A style-based generator architecture for generative adversarial networks. In Proceedings of the IEEE/CVF Conference on Computer Vision and Pattern Recognition (CVPR) (pp. 4401-4410).

[19] Deng, J., Dong, W., Socher, R., Li, L. J., Li, K., & Fei-Fei, L. (2009, June). Imagenet: A large-scale hierarchical image database. In 2009 IEEE Conference on Computer Vision and Pattern Recognition (CVPR) (pp. 248-255).

[20] Karras, T., Aittala, M., Aila, T., & Laine, S. (2022). Elucidating the design space of diffusion-based generative models. Advances in Neural Information Processing Systems (NeurIPS), 35, 26565-26577.

[21] Lou, A., Meng, C., & Ermon, S. (2023). Discrete diffusion modeling by estimating the ratios of the data distribution. International Conference on Machine Learning (ICML), PMLR, 2024.

[22] Zhu, Y., Wang, X., Lathuilière, S., & Kalogeiton, V. (2025). Di $\mathtt {[M]} $ O: Distilling Masked Diffusion Models into One-step Generator. arXiv preprint arXiv:2503.15457.

**Quality:**

3

**Strengths And Weaknesses:**

Pros: This paper studies an important problem (posterior sampling) for the setting of discrete diffusion models, which can be potentially generalized to other problems like inference-time scaling or inference-time alignment. Both theoretical analysis and multiple experiments (along with a wide range of baseline methods) are provided to justify how effective the proposed method is. Moreover, the authors included Python code for the reviewers to test on.

Cons: One of the concerns that the reviewer has is that a few work seem to be missing in the literature review. For instance, for diffusion posterior sampling methods based on the sequential monte carlo (SMC) approach, a lot of recent and concurrent work like [4,5,6,7,8,9,10] probably need to be cited and discussed briefly (Though some of them might have focused on different problem settings like inference-time scaling). Similar issues also exist for other classes of work like guidance-based diffusion posterior sampling and theoretical analysis of discrete diffusion models. Hence, the authors are encouraged to perform a more thorough review of related methods developed in these subfields.

Furthermore, for the theoretical analysis part, the reviewer thinks the assumptions given in Appendix A might be a bit too strong, as they essentially assume that the score function can be ideally learnt with respect to the $l_{\infty}$ norm. Compared to other existing work on the theoretical analysis of discrete diffusion models like [12,13,14,15,16], the assumptions might be too strong. The reviewer will elaborate on such issue in the "Questions" section below.

Moreover, for imaging inverse problems considered in this manuscript, the authors only tested on the MNIST dataset. In contrast, related work like [1,2,17] have tested on large-scale image datasets like FFHQ [18] and ImageNet [19]. The authors are encouraged to test the proposed methodology on these large-scale datasets to justify the effectiveness of the proposed method.

Additionally, several issues and typos also exist regarding the presentation of this paper. For instance, the term $dt$ is missing for almost all integrals of the relative Fisher information appearing in the manuscript (Examples include line 181-182, line 567-570). Also, definition of $\mu_t$ and $\pi_t$ included in line 174-177 seems to be a bit ambiguous. The authors may consider including a similar picture like Figure 3 in [1] to better illustrate the relation between these two processes (or maybe simply referring to Figure 3 in [1]).

---

> ### Author Rebuttal · Authors · 2025-07-31
>
> > 1. More comprehensive literature review of related methods in SMC and inference-time scaling.
>
> We thank the reviewer for helping us improve the related work section. The reviewer mentioned several papers [4-10] that apply SMC methods to the inference-time sampling/scaling of diffusion models. While we implement SVDD [23] and SMC/TDS [24] as two canonical baselines from the class of SMC algorithms, we will cite and discuss all these recent advancements in the paragraph of Sequential Monte Carlo in Section 2.2. Specifically, SMC/TDS [24] can be viewed as a special case of the general FK diffusion sampling framework [4], with the potential function $G\_t$ defined as the difference of value function between two sampling steps, and the proposal function defined as diffusion sampling (vanilla/with classifier guidance).
> To provide a more complete evaluation of SMC-based methods, we further include a FK diffusion sampler [4] with MAX potential function (i.e. the best value along the sampling history), which is found to be slightly more effective in [4]. We use $M=20$ particles for all SMC-based experiments. As we see in the results below, while FK-MAX is slightly better than TDS, our SGDD approach performs significantly better than both SMC-based methods.
>
> | DNA | Pred-Activity (median) | ATAC Acc | Log-likelihood |
> | --  | --| -- | -- |
> | TDS [24] | 4.64 | 45.3% | -257|
> | FK-MAX [4] | 4.90 | 47.1% | -243 |
> | SGDD | 9.14 | 93.0% | -261|
>
> | MNIST-XOR | PSNR | Accuracy |
> | -- | -- | -- |
> | SMC | 10.05 | 27.8% |
> | FK-MAX [4] | 11.20 | 34.7% |
> | SGDD | 20.17 | 91.2% |
>
> > 2. Justifications on our theoretical results.
>
> We appreciate the constructive feedback on our theoretical results. We will make sure to cite the references on the recent theoretical advancement of discrete diffusion models[12-16] in the revised version. Motivated by the reviewer's comment, we improved our assumption by substituting assumptions (i) and (v) with a direct assumption on the score entropy. Specifically, we assume that
> - Score function is well estimated (i.e., the score entropy loss for any distribution is bounded): for any path of distributions $(p\_t)\_{t\in [0,1]}$, it holds that $$\mathbb{E}\_{\mathbf x\_i\sim p\_t} \mathsf{SE}\_{\mathbf Q\_t^{\mathsf{fw}}}(\mathbf x\_i; \theta,t) := \mathbb E\_{\mathbf x\_i\sim p\_t} \left[\sum\_{\mathbf x\_j\neq \mathbf x\_i} K\left(\frac{s\_\theta(\mathbf{x}\_i,t)\_{\mathbf{x}\_j}}{s(\mathbf{x}\_i,t)\_{\mathbf x\_j}}\right) \mathbf Q\_t^{\mathsf{fw}\ [i,j]}s(\mathbf{x}\_i, t)\_{\mathbf{x}\_j}\right] \leq \epsilon,$$where $K(a) := a - 1 - \log a$.
>
> Note that here $p\_t$ is arbitrary and may be different from the path on which the DM was trained on. This is in fact expected and highlights the difference between analyzing the generative process of discrete DMs versus analyzing the process of using discrete DMs for solving inverse problems. The ideal process in the generative process of discrete DMs is the same as the process for training (only time going in different directions). In contrast, when using discrete DMs as priors for solving inverse problems, they conduct inference on arbitrary distributions that may be different from the training distributions, which is a generalization gap that requires stronger condition on the DMs for convergence. In fact, the existing works on SMC assume exact score in the posterior sampling task [10,24].
>
> Given this assumption, we can improve Lemma 3 by directly bounding the Lagrangian term with the score entropy loss. Specifically, we have from Lemma 3 in Appendix A.1 that
> $$
> \mathcal{L}\_{\mathbf{Q}\_t}(\mu\_t, \tilde{\mathbf{Q}}\_t \mu\_t) = \sup\_{\varphi\in \mathcal{C}\_b(\mathcal{X})}\sum\_{\mathbf{x}\_i\neq \mathbf{x}\_j} \left[\mu\_t(\mathbf{x}\_i) \mathbf{Q}\_t^{[j,i]} \left(-\frac{\tilde{\mathbf{Q}}\_t^{[j,i]}}{\mathbf{Q}\_t^{[j,i]}} z\_{ij} - e^{-z\_{ij}}\right) \right] + \sum\_{\mathbf{x}\_i} \mu\_t(\mathbf{x}\_i) (\tilde{\mathbf{Q}}\_t^{[i,i]}-\mathbf{Q}\_t^{[i,i]})
> $$
> where we use the fact that $\mathbf 1^T\tilde{\mathbf{Q}}\_t \mathbf{u} = 0$ for any $\mathbf{u}$ and $z\_{ij} := \varphi(\mathbf{x}\_i) - \varphi(\mathbf{x}\_j)$. Consider the function $g(z) = -uz - e^{-z}$. When $u \geq 0$, this function is maximized at $z = -\log u$. So, $-uz - e^{-z} \leq u\log u - u \leq u\log u$. Setting $u = \tilde{\mathbf{Q}}\_t^{[j,i]}/\mathbf{Q}\_t^{[j,i]} = \frac{s\_\theta(\mathbf{x}\_i, t)\_{\mathbf{x}\_j}}{s(\mathbf{x}\_i, t)\_{\mathbf{x}\_j}}$, we have that
> $$
> \mathcal{L}\_{\mathbf{Q}\_t} (\mu\_t, \tilde{\mathbf{Q}}\_t\mu\_t) \leq \sum\_{\mathbf{x}\_i \neq \mathbf{x}\_j} \left[\mu\_t(\mathbf{x}\_i) \mathbf{Q}\_t^{[j,i]} \frac{s\_\theta(\mathbf{x}\_i, t)\_{\mathbf{x}\_j}}{s(\mathbf{x}\_i, t)\_{\mathbf{x}\_j}} \log \frac{s\_\theta(\mathbf{x}\_i, t)\_{\mathbf{x}\_j}}{s(\mathbf{x}\_i, t)\_{\mathbf{x}\_j}}\right] - \sum\_{\mathbf{x}\_i} \mu\_t(\mathbf{x}\_i) \sum\_{j\neq i} (\tilde{\mathbf{Q}}\_t^{[j,i]} - \mathbf{Q}\_t^{[j,i]}) $$
> $$= \mathbb{E}\_{\mathbf{x}\_i\sim \mu\_t} \sum\_{\mathbf{x}\_j\neq \mathbf{x}\_i} \left[ K\left(\frac{s(\mathbf{x}\_i, t)\_{\mathbf{x}\_j}}{s\_\theta(\mathbf{x}\_i, t)\_{\mathbf{x}\_j}} \right) \mathbf{Q}\_t^{\mathsf{fw}\ [i,j]} s\_\theta(\mathbf{x}\_i, t)\_{\mathbf{x}\_j} \right].
> $$
> Note that the last line is the SE of $s$ with respect to $s\_\theta$. Therefore, by swapping $\mu\_t$ with $\pi\_t$ and $\mathbf{Q}\_t$ with $\tilde{\mathbf{Q}}\_t$, we have
> $$
> \mathcal{L}\_{\tilde{\mathbf{Q}}\_t} (\pi\_t, \mathbf{Q}\_t\pi\_t) \leq \mathbb{E}\_{\mathbf{x}\_i\sim \pi\_t} \mathsf{SE}\_{\mathbf{Q}\_t^{\mathsf{fw}}}(\mathbf{x}\_i; \theta,t) \leq \epsilon.
> $$ Therefore, we can obtain an new version of Theorem 1 that incorporates the SE loss by swapping $\mu\_t$ and $\pi\_t$. We will update the theoretical part accordingly.
>
> Regarding the question on the similarity between FI and score entropy, both of them are built off of Bregman divergence. However, we would like to emphasize that they have different goals: the relative FI in our context is a convergence metric that measures the discrepance between intermediate distributions $\mu\_t$ and $\pi\_t$, while the SE objective measures the difference between the discrete score function and the target score function.
>
> > 3. Limited imaging inverse problems.
>
> We only included binary MNIST data with binary string operations as an example of imaging inverse problem, because the primary goal of this paper is to design diffusion posterior sampling methods for categorical data distributions, while images are inherently continuous and are not the main focus of this paper.
> To demonstrate the scalability of SGDD, we conduct an additional experiment on FFHQ 256 with a vector quantized diffusion model. Specifically, we use a pretrained VQVAE model from [25] and train a discrete diffusion model in the discrete latent space. We test our algorithm on super resolution task (x4). While SVDD fails to solve this high-dimensional task with a complicated forward model that includes a latent space decoder, SGDD is able to reconstruct the underlying images with metrics comparable to the results of the algorithms using continuous diffusion models.
>
> | Super-resolution (x4)| PSNR ($\uparrow$)| LPIPS ($\downarrow$)|
> | --- | --- | --- |
> | SGDD|  $26.13\_{\pm 1.69}$ | $0.278\_{\pm 0.033}$ |
> | SVDD   | $12.08\_{\pm 0.39}$ | $0.594\_{\pm 0.061}$ |
>
> > 4. Typos and ambiguities in notations.
>
> Thank the reviewer for pointing out! We will correct the typos and add an illustration of $\pi\_t$ and $\mu\_t$ in the final version.
>
> > 5. Relation with the Girsanov's Theorem for CTMCs.
>
> We thank the reviewers for providing the references, which we will cite in the revised manuscript. Prompted by the reviewer's suggestion, we verified that invoking the Girsanov's Theorem will lead to the same result. Specifically, using the chain rule for relative entropy and Girsanov's Theorem leads to the following:
>
> $$
> \mathsf{KL}(\mu\_T||\pi\_T) - \mathsf{KL}(\mu\_0||\pi\_0) = \int\_0^T \sum\_{\mathbf{x}\_i\neq \mathbf{x}\_j \in \mathcal{X}} K\left(\frac{\mathbf{Q}\_t^{[j,i]}}{\tilde{\mathbf{Q}}\_t^{[j,i]}}\right)\tilde{\mathbf{Q}}\_t^{[j,i]}\mu\_t(\mathbf{x}\_i)\mathrm{d}t - \int\_0^T \sum\_{\mathbf{x}\_i\neq \mathbf{x}\_j \in \mathcal{X}} K\left(\frac{\mathbf{Q}\_t^{[i,j]}\frac{\pi\_t(\mathbf{x}\_j)}{\pi\_t(\mathbf{x}\_i)}}{\tilde{\mathbf{Q}}\_t^{[i,j]}\frac{\mu\_t(\mathbf{x}\_j)}{\mu\_t(\mathbf{x}\_i)}}\right)\tilde{\mathbf{Q}}\_t^{[i,j]}\mu\_t(\mathbf{x}\_j)\mathrm{d}t. $$
> $$= - \int\_0^T \mathsf{FI}\_{\mathbf{Q}\_t}(\mu\_t||\pi\_t)\mathrm{d}t  - \int\_0^T \sum\_{\mathbf{x}\_i, \mathbf{x}\_j \in \mathcal{X}} \log \left(\frac{\mu\_t(\mathbf{x}\_i)}{\pi\_t(\mathbf{x}\_i)}\right)\left(\mathbf{Q}\_t^{[i,j]} - \tilde{\mathbf{Q}}\_t^{[i,j]}\right)\mu\_t(\mathbf{x}\_j)\mathrm{d}t.$$
>
> Note that taking the time derivative of both sides exactly leads to (21) in Lemma 2. Our calculation is given by a direct calculation of the time derivative of KL divergence along the two Markov processes. We will comment on the derivation using the Girsanov's theorem in the revised manuscript.
>
> > 6. Applying advanced solvers to accelerate inference?
>
> Thanks for the suggestion. SGDD naturally supports the use of more advanced solvers such as [16], as it is agnostic to the choice of discrete diffusion samplers. While we are not able to find the implementation of [16] at this time, we believe using such advanced solvers will further speedup SGDD and we leave this as a possible future extension.
>
> Reference:
>
> [23] Li et al.  Derivative-free guidance in continuous and discrete diffusion models with soft value-based decoding, 2024.
>
> [24] Wu et al. Practical and asymptotically exact conditional sampling in diffusion models, 2024.
>
> [25] Esser et al. "Taming Transformers for High-Resolution Image Synthesis." 2021.

---

> ### Comment · Reviewer_am6s · 2025-08-02
> **Response**
>
> The reviewer would like to thank the authors for their detailed response, which has addressed most of my concern. Therefore, I will raise my score from 2 to 3. However, please ensure that all revisions outlined in the rebuttal above (theory, literature review and extra experiments) are incorporated in the final version of the manuscript.

---

### Decision · Program_Chairs · 2025-09-17

**Decision:**

Accept (poster)

**Comment:**

The paper proposes a posterior sampling algorithm based on split Gibbs sampling for discrete diffusion models. It establishes theoretical convergence guarantees and provides experiments demonstrating the effectiveness of the proposed approach. During the discussion phase, the authors also clarified the reviewers’ questions. Overall, this is a solid contribution that will likely be of broad interest to the NeurIPS community. I therefore recommend acceptance of the paper.